# LEARNING TO SEGMENT FROM NOISY ANNOTATIONS: A SPATIAL CORRECTION APPROACH

**Jiachen Yao[1], Yikai Zhang[2], Songzhu Zheng[1], Mayank Goswami[3], Prateek Prasanna[1], Chao Chen[1]**
[1]Stony Brook University, [2]Morgan Stanley, [3]CUNY Queens College
[1]{jiachen.yao,zheng.songzhu,prateek.prasanna,chao.chen.1}@stonybrook.edu
[2]yikai.zhang@morganstanley.com, [3]mayank.isi@gmail.com

## ABSTRACT

Noisy labels can significantly affect the performance of deep neural networks (DNNs). In medical image segmentation tasks, annotations are error-prone due to the high demand in annotation time and in the annotators' expertise. Existing methods mostly assume noisy labels in different pixels are *i.i.d*. However, segmentation label noise usually has strong spatial correlation and has prominent bias in distribution. In this paper, we propose a novel Markov model for segmentation noisy annotations that encodes both spatial correlation and bias. Further, to mitigate such label noise, we propose a label correction method to recover true label progressively. We provide theoretical guarantees of the correctness of the proposed method. Experiments show that our approach outperforms current state-of-the-art methods on both synthetic and real-world noisy annotations. [1]

## 1 INTRODUCTION

Noisy annotations are inevitable in large scale datasets, and can heavily impair the performance of deep neural networks (DNNs) due to their strong memorization power (Zhang et al., 2016; Arpit et al., 2017). Image segmentation also suffers from the label noise problem. For medical images, segmentation quality is highly dependent on human annotators' expertise and time spent. In practice, medical students and residents in training are often recruited to annotate, potentially introducing errors (Gurari et al., 2015; Kohli et al., 2017). We also note even among experts, there can be poor consensus in terms of objects' location and boundary (Menze et al., 2014; Joskowicz et al., 2018; Zhang et al., 2020a). Furthermore, segmentation annotations require pixel/voxel-level detailed delineations of the objects of interest. Annotating objects involving complex boundaries and structures are especially time-consuming. Thus, errors can naturally be introduced when annotating at scale.

Segmentation is the first step of most analysis pipelines. Inaccurate segmentation can introduce error into measurements such as the morphology, which can be important for downstream diagnosis and prognostic tasks (Wang et al., 2019a; Nafe et al., 2005). Therefore, it is important to develop robust training methods against segmentation label noise. However, despite many existing methods addressing label noise in classification tasks (Patrini et al., 2017; Yu et al., 2019; Zhang & Sabuncu, 2018; Li et al., 2020; Liu et al., 2020; Zhang et al., 2021; Xia et al., 2021), limited progress has been made in the context of image segmentation.

A few existing segmentation label noise approaches (Zhu et al., 2019; Zhang et al., 2020b;a) directly apply methods in classification label noise. However, these methods assume the label noise for each pixel is i.i.d. (independent and identically distributed). This assumption is not realistic in the segmentation context, where annotation is often done by brushes, and error is usually introduced near the boundary of objects. Regions further away from the boundary are less likely to be mislabeled (see Fig. 1c for an illustration). Therefore, in segmentation tasks, label noise of pixels has to be spatially correlated. An i.i.d. label noise will result in unrealistic annotations as in Fig. 1b.

We propose a novel label noise model for segmentation annotations. Our model simulates the real annotation scenario, where an annotator uses a brush to delineate the boundary of an object. The noisy boundary can be considered a random yet continuous distortion of the true boundary. To

---

[1]Codes are available at `https://github.com/michaelofsbu/SpatialCorrection`.

capture this noise behavior, we propose a Markov process model. At each step of the process, two Bernoulli variables are used to control the expansion/shrinkage decision and the spatial-dependent expansion/shrinkage strength along the boundary. This model ensures the noisy label is a continuous distortion of the ground truth label along the boundary, as shown in Fig. 1c. Our model also includes a random flipping noise, which allows random (yet sparse) mislabels to appear even at regions far away from the boundary.

Based on our Markov label noise, we propose a novel algorithm to recover the true labels by removing the bias. Since correcting model bias without any reference is almost impossible (Massart & Nédélec, 2006), our algorithm requires a clean validation set, i.e., a set of well-curated annotations, to estimate and correct the bias introduced due to label noise. We prove theoretically that only a small amount of validation data are needed to fully correct the bias and clean the noise. Empirically, we show that a single validation image annotation is enough for the bias correction; this is quite reasonable in practice. Furthermore, we generalize our algorithm to an iterative method that repeatedly trains a segmentation model and corrects labels,

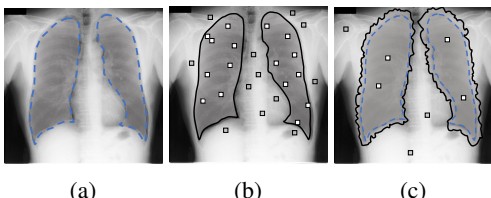

(a)     (b)     (c)

Figure 1: (a) Original image with true segmentation boundary (blue dash line). (b) Classification label noise model in segmentation context is unrealistic, where the label noise (small squares) spread allover the mask. (c) A realistic segmentation noise generated by our noise model. The noise is mostly about distortions of the boundary. A few random flippings appear at the interior/exterior.

until convergence. Since our algorithm, called *Spatial Correction (SC)*, is separate from the DNN training process, it is agnostic to the backbone DNN architecture, and can be combined with any segmentation model. On a variety of benchmarks, our method demonstrates superior performance over different state-of-the-art (SOTA) baselines. To summarize, our contribution is three-folds.

- We propose a Markov model for segmentation label noise. To the best of our knowledge, this is the first noise model that is tailored for segmentation task and considers spatial correlation.
- We propose an algorithm to correct the Markov label noise. Although a validation set is required to combat bias, we prove that the algorithm only needs a small amount of validation data to fully recover the clean labels.
- We extend the algorithm to an iterative approach (SC) that can handle more general label noise in various benchmarks and we show that it outperforms SOTA baselines.

## 2 RELATED WORK

Methods in classification label noise can be categorized into two classes, i.e., model re-calibration and data re-calibration. Model re-calibration methods focus on training a robust network using given noisy labels. Some estimate a noise matrix through special designs of network architecture (Sukhbaatar et al., 2015; Goldberger & Ben-Reuven, 2017) or loss functions (Patrini et al., 2017; Hendrycks et al., 2018). Some design loss functions that are robust to label noise (Zhang & Sabuncu, 2018; Wang et al., 2019b; Liu & Guo, 2020; Lyu & Tsang, 2020; Ma et al., 2020). For example, *generalized cross entropy* (GCE) (Zhang & Sabuncu, 2018) and *symmetric cross entropy* (SCE) (Wang et al., 2019b) combine both the robustness of mean absolute error and classification strength of cross entropy loss. Other methods (Xia et al., 2021; Liu et al., 2020; Wei et al., 2021) add a regularization term to prevent the network from overfitting to noisy labels. Model re-calibration methods usually have strong assumptions and have limited performance when the noise rate is high. Data re-calibration methods achieve SOTA performance by either selecting trustworthy data or correcting labels that are suspected to be noise. Methods like Co-teaching (Han et al., 2018) and (Jiang et al., 2018; Yu et al., 2019) filter out noisy labels and train the network only on clean samples. Most recently, Tanaka et al. (2018); Zheng et al. (2020); Zhang et al. (2021) propose methods that can correct noisy labels using network predictions. Li et al. (2020) extends these methods by maintaining two networks and relabeling each data with a linear combination of the original label and the confidence of the peer network that takes augmented input.

**Training Segmentation Models with Label Noise.** Most existing methods adapt methods for classification to the segmentation task. Zhu et al. (2019) utilize the sample re-weighting technique to train a robust model by adding more weights on reliable samples. Zhang et al. (2020c) extend Co-teaching (Han et al., 2018) to Tri-teaching. Three networks are trained jointly, and each pair of

networks alternatively select informative samples for the third network learning, according to the consensus and difference between their predictions. Zhang et al. (2020b) corrects pixel-wise label noise directly using network prediction, while Li et al. (2021) is based on superpixels. And Liu et al. (2022) apply early learning techniques into segmentation. All these methods fail to address the unique challenges in segmentation label noise (see Section A.3), namely, the spatial correlation and bias. Since segmentation label noise concentrates around the boundary, methods taking every pixel independently will easily fail.

## 3 METHOD

We start by introducing our main intuition about each subsection before we discuss its details. In Section 3.1, we aim at modelling the noise due to inexact annotations. When an annotator segments an image by marking the boundary, the error annotation process resembles a random distortion around the true boundary. In this sense, the noisy segmentation boundary can be obtained by randomly distorting the true segmentation boundary. We model the random distortion with a Markov process. Each Markov step is controlled by two parameters $\theta_1$ and $\theta_2$. $\theta_1$ controls the probability of expansion/shrinkage. It has probability $\theta_1$ to move towards exterior and $1 - \theta_1$ to move towards interior. $\theta_2$ represents of the probability of marching, i.e., a point on the boundary have probability $\theta_2$ to take a step and $1 - \theta_2$ to halt. Fig. 3 illustrates such a process. We start with the true label, go through two expansion steps and one shrinkage step. At each step, we mark the flipped boundary pixels. If $\theta_1 \neq 0.5$, the random distortion will have a preference to expansion/shrinkage. This will result in a bias in the expected state. A DNN trained with such label noise be inevitably affected by the bias. Theoretically, this bias is challenging to be corrected by existing label noise methods, and in general by any bias-agnostic methods (Massart & Nédélec, 2006). Therefore, we require a reasonably small validation set to remove the bias.

In Section 3.2, we propose a provably-correct algorithm to correct the noisy labels by removing this bias. We start by $T = 1$. Since every pixel on the foreground/background boundary has the same probability to be flipped into noisy label, the expected state, i.e. the expectation of the Markov process, only has three cases, taking one step outside, taking one step inside, or staying unchanged. This indicates the relationship between the expected state and the true label can be linearized. We prove this using *signed distance representation* in Lemma 1. If the expected state for each image is given, with only one corresponding true label, we can recover the bias in the linear equation. However, in practice, the DNN is learned to predict the expected state, and there will be an approximation error. Therefore, more validation data may be required to get a precise estimation. In Theorem 1 we prove that with a fixed error and confidence level, the necessary validation set size is only $O(1)$.

The algorithm in Section 3.2 is designed for our Markov noise, where each point's moving probability only depends on its relative position on the boundary. In real-world, this probability can also be feature-dependent. To combat more general label noise, in Section 3.3, we extend our algorithm to correct labels iteratively based on logits. In practice, logits are the network outputs before sigmoid. Unlike distance function, logits contain feature information. A larger absolute logit value means more confident the prediction can be. Subtracting the bias from logits can move boundary points according to features, i.e., regions with large confidence move less than regions with small confidence. After correcting noisy labels, we retrain the DNN with new labels and do this iteratively until the estimated bias is small enough. We summarize our framework in Figure 2.

**Notations.** We assume a 2D input image $\boldsymbol{X} \in \mathbb{R}^{H \times W \times C}$ with *height $H$*, *width $W$*, and *channel $C$*, although our algorithm naturally generalizes to 3D images. $\boldsymbol{Y} = \{0, 1\}^{H \times W \times L}$ is the underlying true segmentation mask with $L$ classes in a one-hot manner. When training a segmentation model with label noise, we are provided with a noisy training dataset $\tilde{\mathcal{D}} = \{(\boldsymbol{X}_n, \tilde{\boldsymbol{Y}}_n)\}_{n=1}^N$. Here all noisy masks are sampled from the same distribution $P(\tilde{\boldsymbol{Y}}|\boldsymbol{X})$. $P(\boldsymbol{Y}|\boldsymbol{X})$ denotes the true label distribution. We use subscript $s \in I$ to represent pixel index, where $I = \{(i,j)|1 \leq i \leq H, 1 \leq j \leq W\}$ is the index set. The scalar form $X_s, Y_s$ denote the pixel value of index $s$ and its label. For the rest of the paper, we assume a binary segmentation task with Foreground (FG) and Background (BG) labels. Since $\boldsymbol{Y}$ is defined in one-hot manner, our formula and algorithm can be generalized to multi-class segmentation easily.

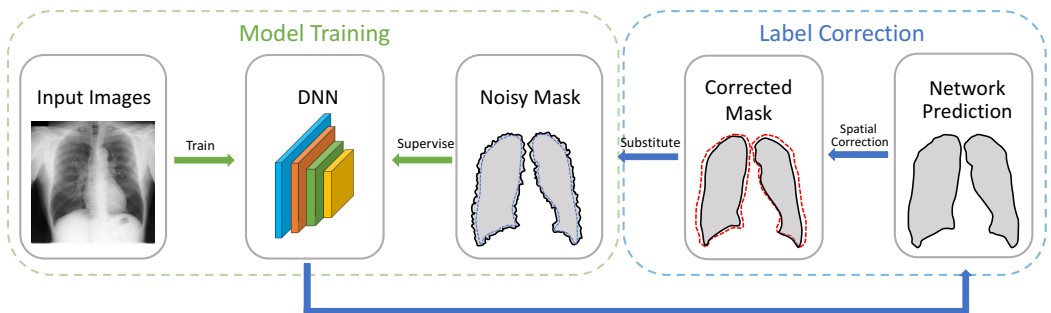

Figure 2: Framework of our method. We train a DNN using noisy labels. The learned DNN prediction boundary (red dashed line) is corrected to the new boundary (black solid line). We use corrected labels to re-train the network. The iterative algorithm can correct the noisy predictions to true labels progressively.

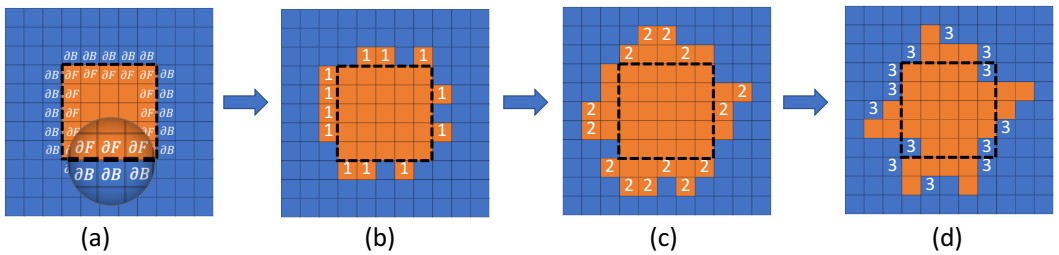

<p style="text-align: center;">(a)       (b)       (c)       (d)</p>

Figure 3: Illustration of a 3-step Markov process. (a) The true label mask, where red pixels are foreground and blue pixels are background. We mark background boundary pixels and foreground boundary pixels as $\partial B$ and $\partial F$, respectively. (b) An expansion step. Pixels marked as '1' have been flipped into foreground. The flipped pixels are randomly chosen with probability $\theta_2$ from the $\partial B$ pixels in (a). (c) Another expansion step by flipping pixels marked as '2' to foreground. (d) A shrinkage step. Pixels marked as '3' were foreground in (c) but are flipped into background in (d).

## 3.1 Modeling Label Noise as A Markov Process

For an input image $X$, we denote the clean and noisy masks $Y$ and $\tilde{Y} \in \{0,1\}^{H \times W}$. The finite Markov process is denoted as $M_\epsilon(T, \theta_1, \theta_2)$, with $T$ denoting the number of steps. $\theta_1$ and $\theta_2$ are two Bernoulli parameters denoting the annotation preference and annotation variance, respectively. To further enhance the modeling ability, we also introduce random flipping noise at regions far away from the boundary into our model. This is achieved by adding a matrix-valued random noise $\epsilon \sim \{\text{Bernoulli}(\theta_3)\}^{H \times W}$ into the final step. The formal definition of $M_\epsilon(T, \theta_1, \theta_2)$ is in Definition 1.

Denote by $F$, $B$ the foreground and background masks, i.e. $F = Y$ and $B = 1 - Y$. We define the boundary operator $\partial \cdot$ of $F$ and $B$. Let $\partial F = \mathbf{1}_{\{s|F_s=1, \exists r \in N_s, B_r=1\}}$ be the boundary mask of $F$, i.e. foreground pixels adjacent to $B$ holding value 1, otherwise 0. $N_s$ is the four-neighbor of index $s$. Similarly, let $\partial B$ be the boundary mask of $B$. Note that sets $\{\partial F = 1\}$ and $\{\partial B = 1\}$ are on the opposite sides of the boundary, as shown in Fig. 3(a). For simplification, we will abuse the notation $\partial B$ for both a matrix and a set of indices $\{\partial B = 1\}$. Readers will notice the difference easily. The Markov process at the $t$-th step generates the $t$-th noisy label $\tilde{Y}^{(t)}$ based on the $(t-1)$-th noisy label $\tilde{Y}^{(t-1)}$. Denote by $F^{(t-1)}$ and $B^{(t-1)}$ the foreground and background masks of $\tilde{Y}^{(t-1)}$.

**Definition 1** (Markov Label Noise). *Let $\tilde{Y}^{(0)} = Y$. For $t = 0, 1, ..., T-1$, let $z_1^{(t)} \overset{i.i.d.}{\sim}$ Bernoulli($\theta_1$), and $Z_2^{(t)} \overset{i.i.d.}{\sim} \{\text{Bernoulli}(\theta_2)\}^{H \times W}$.*

$$\tilde{Y}^{(t+1)} = \tilde{Y}^{(t)} + z_1^{(t)} Z_2^{(t)} \odot \partial B^{(t)} + (z_1^{(t)} - 1) Z_2^{(t)} \odot \partial F^{(t)}, \tag{1}$$

*The final output noisy label is $\tilde{Y} = \tilde{Y}^{(T)} + \epsilon \odot \text{Sign}$, where $\text{Sign} = B^{(T)} \odot B - F^{(T)} \odot F$. The process is denoted by $M_\epsilon(T, \theta_1, \theta_2)$.*

The random variable $z_1^{(t)}$ determines whether the noise is obtained by expanding or shrinking the boundary in each step $t$. If expansion, $z_1^{(t)} = 1$, the error is taken by flipping the background

boundary pixels $\partial \boldsymbol{B}^{(t-1)}$ with probability $\theta_2$. This is encoded in the second Bernoulli $\boldsymbol{Z}_2^{(t)} \odot \partial \boldsymbol{B}^{(t)}$, where $\partial \boldsymbol{B}^{(t)}$ is an indication matrix to restrict the flipping only happens among background boundary pixels. Similarly, if shrinkage, $z_1^{(t)} = 0$, a pixel in $\partial \boldsymbol{F}^{(t-1)}$ has probability $\theta_2$ to be flipped into background. Fig. 3 shows an example with 3 steps, corresponding to expansion, expansion and shrinkage.

## 3.2 LABEL CORRECTION BY REMOVING BIAS

Suppose the Markov noise has underlying posterior $P(\tilde{\boldsymbol{Y}}|\boldsymbol{X})$, which is impossible to get a general explicit form due to the progressive spatial dependency. However, we can study the Bayes classifier $\tilde{c}(\boldsymbol{X}) = \arg\max_{\tilde{\boldsymbol{Y}}} P(\tilde{\boldsymbol{Y}}|\boldsymbol{X})$. For a fixed image $\boldsymbol{X}$, we take $\tilde{\boldsymbol{Y}} \sim P(\tilde{\boldsymbol{Y}}|\boldsymbol{X})$ as a random variable. Then $\mathbb{E}[\tilde{\boldsymbol{Y}}]$ and $\tilde{c}(\boldsymbol{X})$ have,

$$\tilde{c}(\boldsymbol{X}) = [\mathbb{E}[\tilde{\boldsymbol{Y}}]]_{\geq 0.5}, \tag{2}$$

where $[\mathbb{E}[\tilde{\boldsymbol{Y}}]]_{\geq 0.5}$ means for each index $s$, $[\mathbb{E}[\tilde{\boldsymbol{Y}}]]_{\geq 0.5}(s) = 1$, if $[\mathbb{E}[\tilde{Y}_s]] \geq 0.5$, otherwise, 0. Note that $\mathbb{E}[\tilde{\boldsymbol{Y}}]$ is a probability map while $[\mathbb{E}[\tilde{\boldsymbol{Y}}]]_{\geq 0.5}$ is its (Bayes) prediction. Therefore, to recover the true Bayes classifier $c(\boldsymbol{X})$ is to build an equation between $\mathbb{E}[\tilde{\boldsymbol{Y}}]$ and $\mathbb{E}[\boldsymbol{Y}]$. we iterate our Markov model Equation 1 over $t = 0, ..., T-1$, plus the random flipping term and take expectation on both sides,

$$\mathbb{E}[\tilde{\boldsymbol{Y}}] = \mathbb{E}[\boldsymbol{Y}] + \theta_1\theta_2 \sum_{t=0}^{T-1} \mathbb{E}[\partial \boldsymbol{B}^{(t)}] + (\theta_1 - 1)\theta_2 \sum_{t=0}^{T-1} \mathbb{E}[\partial \boldsymbol{F}^{(t)}] + \theta_3\mathbb{E}[\textbf{Sign}]. \tag{3}$$

Equation 3 defines the bias between $\mathbb{E}[\boldsymbol{Y}]$ and $\mathbb{E}[\tilde{\boldsymbol{Y}}]$. But the ambiguous representation of $\mathbb{E}[\partial \boldsymbol{B}^{(t)}]$ makes it difficult to simplify the bias. In order to analyze it, we need to transform the explicit boundary representation $\partial \boldsymbol{B}$ into an *implicit function*. For a domain $\Omega$, we define its interface (boundary) as $\partial\Omega$, its interior and exterior as $\Omega^-$ and $\Omega^+$, respectively. An implicit interface representation defines the interface as the isocontour of some function $\phi(\cdot)$. Typically, the interface is defined as $\partial\Omega = \{s \in \Omega | \phi(s) = 0\}$.

A straightforward implicit function is the *signed distance function*. Following Osher & Fedkiw (2003), a *distance function* $d(s)$ is defined as $d(s) = \min_{t \in \partial\Omega}(|s - t|)$, implying that $d(s) = 0$ on $s \in \partial\Omega$. A *signed distance function* is an implicit function $\phi$ defined by $|\phi(s)| = d(s)$ for all $s \in \Omega$, s.t. $\phi(s) = d(s) = 0$ for $s \in \partial\Omega$, $\phi(s) = d(s)$ for $s \in \Omega^+$, and $\phi(s) = -d(s)$ for $s \in \Omega^-$. For a domain of grid points, we define the distance function $d(s)$ by the shortest path length from $s$ to $\partial\Omega$. But note that $\partial\Omega$ does not exist among image indices. It lies between $\partial \boldsymbol{B}$ and $\partial \boldsymbol{F}$. The corresponding definition for image indices is as follows.

**Definition 2** (Signed Distance Function). *For an Image $\boldsymbol{X}$ with index set $I$, a graph $G(S, \mathcal{E})$ is constructed by $S = I$ and $\mathcal{E} = \{s \to t : s \in I, t \in N_s\}$, where $N_s$ is the four-neighbor of index $s$. Then the distance $d(s, t)$ is defined by the length of the shortest path between $s$ and $t$ in graph $G$. The distance function is defined by*

$$d(s) = \begin{cases} \min_{t \in \partial \boldsymbol{B}} d(s, t) + 1, & \text{if } s \in \boldsymbol{B}, \\ \min_{t \in \partial \boldsymbol{F}} d(s, t) + 1, & \text{otherwise.} \end{cases} \tag{4}$$

*The signed distance function $\phi(s)$ is then defined by $\phi(s) = d(s)$, if $s \in B$. Otherwise, $\phi(s) = -d(s)$.*

We start from the case $T = 1$. With the signed distance representation, we have the following lemma.

**Lemma 1.** *$\phi$ is the signed distance function defined for $c(\boldsymbol{X})$. If $\theta_3 \ll 0.5$, then*

$$\tilde{\phi} = \begin{cases} \phi - [\theta_1\theta_2]_{\geq 0.5} - 1, & \text{if } \theta_1 \geq 0.5, \\ \phi + [1 + \theta_1\theta_2 - \theta_2]_{<0.5} + 1 & \text{otherwise.} \end{cases} \tag{5}$$

*Here $\tilde{\phi}$ is the signed distance function of $\tilde{c}(\boldsymbol{X})$ when $T = 1$.*

With the *signed distance function*, we can explicitly define the bias by $\Delta = \frac{1}{|I|} \sum_{s \in I}(\tilde{\phi}_s - \phi_s)$, where $|I|$ is the cardinality of the index set, also the image size. If $\tilde{c}(\boldsymbol{X})$ is perfectly learned, the

difference between $\tilde{\phi}$ and $\phi$, i.e. $\Delta$ can be estimated with a small clean validation set, even if there is only one image. Then the original function $\phi$ can be recovered among training set by $\phi = \tilde{\phi} + \Delta$. Then $c(\boldsymbol{X})$ can be simply obtained by $[\phi]_{\leq 0}$. However, in practice, the classifier learned, denoted by $\hat{c}(\boldsymbol{X})$, is different from the noisy Bayes optimal $\tilde{c}(\boldsymbol{X})$. This will lead an error to $\hat{\phi}$, *signed distance function* of $\hat{c}(\boldsymbol{X})$, for which more validation data may be required. The naive algorithm is,

1. *Bias estimation.* Train a DNN with noisy labels. Given a clean validation set $\{\mathbf{x}_v, \mathbf{y}_v\}_{v=1}^V$, the bias is estimated by

$$\hat{\Delta} = \frac{1}{V} \sum_{v=1}^V \frac{1}{|I|} \sum_{s \in I} (\hat{\phi}_s^v - \phi_s^v). \tag{6}$$

2. *Label correction.* Correct training labels by $c'(\boldsymbol{X}) = [\phi']_{\leq 0}$, where $\phi' = \hat{\phi} - \hat{\Delta}$. Then retrain the network using corrected labels.

In Theorem 1 we prove that with a small validation size, the above algorithm can recover the true label with bounded error.

**Theorem 1.** *If* $\exists\ \varepsilon_0 > 0,\ \varepsilon_1 > 0,\ s.t.\ \mathbb{E}_{\boldsymbol{X}} \left[ \sup_s |\hat{\phi}(X_s) - \tilde{\phi}(X_s)| \right] \leq \varepsilon_0$ *and* $\sup_{\boldsymbol{X}} \left[ \sup_s |\hat{\phi}(X_s) - \tilde{\phi}(X_s)| \right] \leq \varepsilon_1$ *hold for the learned classifier* $\hat{c}(\boldsymbol{X})$, *then* $\forall \varepsilon > \varepsilon_0$ *and for a fixed confidence level* $0 < \alpha \leq 1$, *with*

$$V \geq \frac{\varepsilon_1^2}{2(\varepsilon - \varepsilon_0)^2} \log \left( \frac{2|I|}{\alpha} \right) \tag{7}$$

*number of clean samples* $\{\mathbf{x}_v, \mathbf{y}_v\}_{v=1}^V$, $\phi'$ *can be recovered within* $\varepsilon + \varepsilon_0$ *error with probability at least* $1 - \alpha$, *i.e.* $P(\mathbb{E}_{\boldsymbol{X}} [\sup_s |\phi'(X_s) - \phi(X_s)|] \leq \varepsilon + \varepsilon_0) \geq 1 - \alpha$.

Proofs of Lemma 1 and Theorem 1 are provided in Appendix. The theorem shows that for a fixed error level $\varepsilon$ and a fixed confidence level $\alpha$, the validation size $V$ required is a constant, logarithm of the image size. Note that $\varepsilon_0$ is the mean model error among images and $\varepsilon_1$ is the supremum model error among images. It holds naturally that $\varepsilon_0 \leq \varepsilon_1$. $\varepsilon_1$ is to constrain the model prediction variance on different images. If $\varepsilon_0 = \varepsilon_1$, the model prediction quality is similar over images. Even in the worst case, where model cannot predict any right label, $\varepsilon_1$ is still bounded by the image size $|I|$.

## 3.3 ITERATIVE LABEL CORRECTION

In this section, we extend our algorithm to more general label noise. In our Markov model, all points on the boundary are moving with the same probability along the normal direction of a *signed distance function*. The boundary in real-world noise, however, could have feature-dependent moving probability. We achieve this by using *logit function representation*. For an image $\boldsymbol{X}$, its *logit function* $f(\boldsymbol{X})$ is defined by $\mathbb{E}\boldsymbol{Y} = \sigma \circ f(\boldsymbol{X})$, where $\sigma(\cdot)$ is a *sigmoid function* and $\boldsymbol{Y}$ is the segmentation label. Note that $f(\boldsymbol{X})$ is positive in interior $\Omega^-$ and negative in $\Omega^+$, so the *implicit function* is defined by $-f(\boldsymbol{X})$. The bias estimation step remains the same. But at the label correction step, we apply the bias on $\hat{f}(\boldsymbol{X})$ instead of $\hat{\phi}$. In practice, $\hat{f}(\boldsymbol{X})$ is the model output before the sigmoid function. The label correction step by *logit function* is

$$f'(\boldsymbol{X}) = \hat{f}(\boldsymbol{X}) + \lambda \exp \left[ -\frac{\hat{\phi}^2}{2(\gamma\hat{\Delta})^2} \right]. \tag{8}$$

$\lambda$ is the bias adapted to logit function and the exponential term is a decay function to constrain the bias around $\hat{\phi} = 0$. And the decay factor is $\gamma\hat{\Delta}$, where $0 < \gamma \leq 1$ is a hyper-parameter, usually set to be 1. The bias $\lambda$ is defined by $\lambda = \inf \hat{f}|_{0 \leq \hat{\phi}_s \leq \hat{\Delta}}$, if $\hat{\Delta} > 0$, and $\lambda = \sup \hat{f}|_{\hat{\Delta} \leq \hat{\phi}_s \leq 0}$, if $\hat{\Delta} < 0$. $f|_\Omega$ is function $f$ restricted on domain $\Omega$. Although the bias is decayed with the same scale over the distance transform, the gradient of logit function, however, varies along the normal direction. Therefore, Equation 8 actually moves points on the interface with different degrees. Points with smaller absolute gradient moves more. Since the algorithm corrects noisy labels spatially correlated to the boundary, we refer to it as *Spatial Correction (SC)*. We present our SC as an iterative approach in Algorithm 1. In practice, the hyper-parameter $\gamma$ is usually set to be 1, and the algorithm can terminate after only 1 iteration.

---

**Algorithm 1** Spatial Correction

---

**Input:** Noisy training dataset $\tilde{\mathcal{D}}$, a small clean validation dataset $\mathcal{V}$, and a hyper-parameter $\gamma$.
**Output:** A robust DNN $\hat{f}$ trained with denoised dataset.
    Train a DNN $\hat{f}$ with $\tilde{\mathcal{D}}$.
    Predict labels on $\mathcal{V}$ using $\hat{f}$, and estimate $\hat{\Delta}$ by Equation 6.
    **while** $|\hat{\Delta}| \geq 1$ **do**
        Replace training labels with $[f'(\boldsymbol{X})]_{\geq 0}$, where $f'(\boldsymbol{X})$ is calculated by Equation 8.
        Re-train $\hat{f}$ with corrected labels.
        Predict labels on $\mathcal{V}$ using new $\hat{f}$, and estimate $\hat{\Delta}$ by Equation 6.
    **end while**
    **return** $\hat{f}$.

---

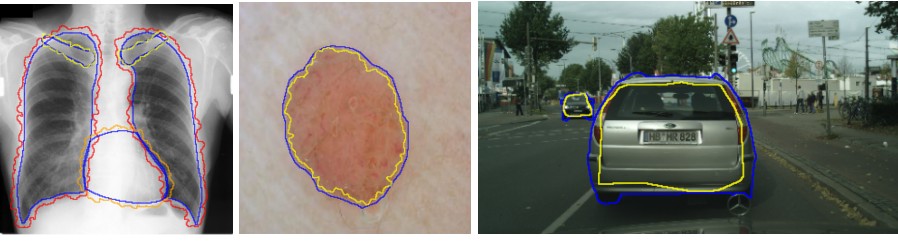

(a) *JSRT $S_E$.*          (b) *ISIC 2017 $S_S$.*          (c) *Cityscapes.*

Figure 4: Examples of synthetic and real-world noise. In each image, blue line is the true segmentation boundary, and all other colors are corresponding noisy boundaries. We removed the random flipping noise in visualization to focus on the boundary.

## 4 EXPERIMENTS

In this section, we provide an extensive evaluation of our approach with multiple datasets and noise settings. We also show in ablation studies, that our method is robust to high noise level and the required clean validation set size can be extremely small.

### 4.1 DATASETS AND IMPLEMENTATION DETAILS

**Synthetic noise settings.** We use three public medical image datasets, *JSRT* dataset (Shiraishi et al., 2000), *ISIC 2017* dataset (Codella et al., 2017), and *Brats 2020* dataset (Menze et al., 2015; Bakas et al., 2017b; 2018; 2017a). *JSRT* contains 247 images for chest CT and three types of organ structures: lung, heart and clavicle, all with clean ground-truth labels (Juhász et al., 2010). We randomly split the data into training (148 images), validation (24 images), and test (75 images) subsets. *ISIC 2017* is a skin lesion segmentation dataset with 2000 training, 150 validation, 600 test images. Following standard practice Zhu et al. (2019); Zhang et al. (2020b); Li et al. (2021), we resize all images in both datasets to $256 \times 256$ in resolution. *Brats 2020* is a brain tumor 3D segmentation dataset with 369 training volumes. Since only training labels are accessible, we randomly split these 369 volumes into training (200 volumes), validation (19 volumes) and test (150 volumes) subsets. We resize all volumes to $64 \times 128 \times 128$ in resolution.

For each of these three datasets, we use three noise settings, denoted by $S_E$, $S_S$ and $S_M$. $S_E$ and $S_S$ are two settings synthesized by our Markov process with $\theta_1 > 0.5$ (expansion) and $\theta_1 < 0.5$ (shrinkage), respectively. Figure 4 shows examples of our synthesized label noise. We also include the mix of random dilation and erosion noise $S_M$ used by previous work (Zhu et al., 2019; Zhang et al., 2020b;a). This is achieved by randomly dilate or erode a mask with a number of pixels. Note that our Markov label noise can theoretically include this type of noise by setting $\theta_1 = 0.5$. Detailed parameters for these settings are provided in the Appendix.

We use a simple U-Net as our backbone network for *JSRT* and *ISIC 2017* datasets and a 3D-UNet for *Brats 2020* dataset. The hyper-parameter $\gamma$ is set to be 1 and total iteration is 1.

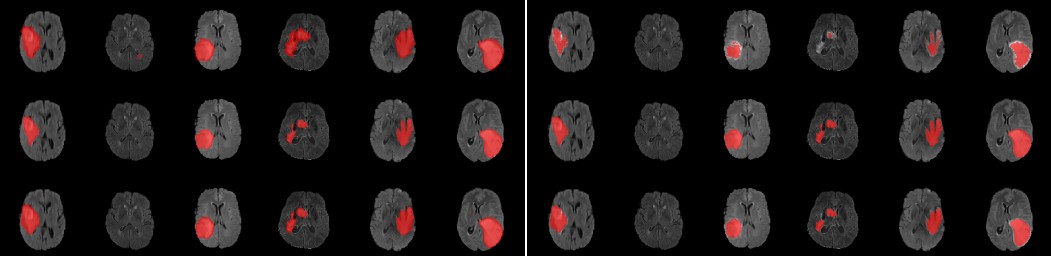

Figure 5: Label correction results for *Brats 2020* datasets with $S_E$ (Left) and $S_S$ (Right) settings. From top to bottom, the rows are sample slices with noisy masks, with true masks, and with corrected masks, respectively.

**Real-world label noise.** To evaluate with real-world label noise is challenging. We are not aware of any public medical image segmentation dataset that has both true labels and noisy labels from human annotators. Therefore, we use a multi-annotator dataset, *LIDC-IDRI* dataset (Armato III et al., 2015; Armato et al., 2011; Clark et al., 2013), and the coarse segmentation in a vision dataset, *Cityscapes* (Cordts et al., 2016). The *LIDC-IDRI* dataset consists of 1018 3D thorax CT scans where four radiologists have annotated multiple lung nodules in each scan. The dataset was annotated by 12 radiologists, and it is not possible to match an annotation to an expert. We use the majority voting as the true labels and the union of four annotations as noisy labels. We process and split the data exactly the same way as Kohl et al. (2018). *Cityscapes* dataset contains 5000 finely annotated images along with a coarse segmentation by human annotators that we use as the "noisy label". We only focus on the 'car' class because (1) cars are popular objects and are frequently included in images; (2) the coarse annotation of cars is very similar to noisy annotation in medical imaging – they are reasonable distortions of the clean label without changing the topology. See Figure 4c for an example. The detailed settings of *LIDC-IDRI* and *Cityscapes* can be found in Appendix A.2.1.

**Baselines.** We compare the proposed SC with SOTA learning-with-label-noise methods from both classification (GCE (Zhang & Sabuncu, 2018), SCE (Wang et al., 2019b), CT+ (Yu et al., 2019), ELR (Liu et al., 2020)), CDR (Xia et al., 2021) and segmentation contexts (QAM (Zhu et al., 2019), CLE (Zhang et al., 2020b)). Technical details of these baseline methods are provided in Appendix A.2.2. Our method requires a small clean validation set whereas most baselines do not. For a fair comparison, we also use the clean validation set to strengthen the baselines. In particular, we pretrain the baselines using the clean validation dataset, and then add the validation images and their clean labels into the training set. Note that our method (SC) is only trained on the original noisy trianing set; it only uses the validation set to estimate bias.

## 4.2 RESULTS

Table 1 shows the segmentation results of different methods with synthetic noisy label settings on *JSRT* , *ISIC 2017* and *Brats 2020* dataset. Note that QAM cannot be applied to *Brats 2020* dataset because their network is designed for 2D only. We compare DICE score (DSC) on testing sets (against the clean labels). For each setting, we train 5 different models, and report the mean DSC and standard deviation. In $S_E$ and $S_S$, where biases show up in noisy labels, the proposed method outperforms the baselines by a big leap in total case. The compared methods, however, only work when little bias is included, like $S_M$. $S_M$ is equivalent to setting $\theta_1 = 0.5$ in our Markov model, resulting in $\Delta = 0$. We also test the proposed method on real-world label noise, results shows in Table 2. Figure 5 shows examples of label correction results. We provide more qualitative results in the Appendix A.4.

## 4.3 ABLATION STUDY

**Increasing Noise Level.** Our proposed method is robust even when the noise level is high. In Figure 6a, we compare the prposed SC with SCE and ELR. We increase the synthetic noise level on *JSRT* dataset by increasing the step $T$ in our Markov model, while keep the same $\theta_1$ and $\theta_2$ (details and illustrations in the supplementary material). Results show our method is still robust even under extreme noise level, while the performance of GCE and ELR drops rapidly as the noise level increases.

Table 1: Mean DSC (in percent) and standard deviation for five models trained on three noisy settings. Method with best mean DSC is highlighted for each noise setting.

| | Method | JSRT | | | | ISIC 2017 | Brats 2020 |
| | | Lung | Heart | Clavicle | Total | | |
|---|---|---|---|---|---|---|---|
| $S_E$ | GCE | $79.43 \pm 0.17$ | $71.70 \pm 0.36$ | $58.56 \pm 0.14$ | $69.90 \pm 0.12$ | $73.12 \pm 0.27$ | $68.07 \pm 0.78$ |
| | SCE | $81.03 \pm 1.98$ | $79.24 \pm 2.90$ | $73.38 \pm 1.98$ | $77.88 \pm 1.45$ | $74.06 \pm 0.92$ | $68.04 \pm 1.71$ |
| | CT+ | $82.08 \pm 0.06$ | $66.57 \pm 0.08$ | $44.63 \pm 0.10$ | $64.43 \pm 0.07$ | $68.65 \pm 0.20$ | $70.69 \pm 0.72$ |
| | ELR | $79.45 \pm 0.26$ | $79.69 \pm 0.56$ | $\mathbf{79.37 \pm 0.18}$ | $79.51 \pm 0.16$ | $72.64 \pm 0.99$ | $67.56 \pm 0.82$ |
| | CDR | $78.77 \pm 0.36$ | $73.64 \pm 1.03$ | $63.77 \pm 0.93$ | $72.06 \pm 0.50$ | $72.76 \pm 0.95$ | $66.95 \pm 2.23$ |
| | QAM | $78.54 \pm 0.45$ | $78.63 \pm 0.20$ | $78.71 \pm 0.14$ | $78.63 \pm 0.14$ | $71.32 \pm 0.45$ | $--$ |
| | CLE | $79.49 \pm 0.83$ | $73.19 \pm 0.65$ | $45.75 \pm 2.48$ | $66.14 \pm 0.78$ | $72.95 \pm 1.21$ | $65.93 \pm 1.39$ |
| | SC(Ours) | $\mathbf{91.62 \pm 2.85}$ | $\mathbf{89.19 \pm 0.96}$ | $78.11 \pm 1.35$ | $\mathbf{86.04 \pm 1.76}$ | $\mathbf{80.64 \pm 0.63}$ | $\mathbf{77.78 \pm 0.85}$ |
| $S_S$ | GCE | $88.31 \pm 0.40$ | $90.84 \pm 0.17$ | $71.53 \pm 0.36$ | $83.56 \pm 0.19$ | $52.44 \pm 3.47$ | $47.65 \pm 1.42$ |
| | SCE | $68.04 \pm 1.71$ | $75.83 \pm 1.59$ | $\mathbf{82.49 \pm 0.30}$ | $78.67 \pm 1.38$ | $53.03 \pm 3.34$ | $47.01 \pm 2.11$ |
| | CT+ | $93.55 \pm 0.11$ | $84.11 \pm 0.14$ | $53.86 \pm 0.17$ | $77.18 \pm 0.11$ | $67.55 \pm 0.53$ | $71.69 \pm 0.01$ |
| | ELR | $74.15 \pm 0.95$ | $71.15 \pm 0.41$ | $71.76 \pm 0.25$ | $72.35 \pm 0.37$ | $50.21 \pm 3.41$ | $52.36 \pm 5.48$ |
| | CDR | $83.20 \pm 2.06$ | $83.51 \pm 2.13$ | $75.53 \pm 1.28$ | $80.74 \pm 1.04$ | $50.87 \pm 1.56$ | $51.51 \pm 7.98$ |
| | QAM | $72.76 \pm 0.91$ | $68.71 \pm 1.51$ | $70.68 \pm 0.80$ | $70.72 \pm 0.91$ | $52.95 \pm 3.10$ | $--$ |
| | CLE | $81.97 \pm 1.45$ | $83.86 \pm 1.30$ | $50.56 \pm 2.83$ | $72.13 \pm 1.30$ | $54.76 \pm 1.15$ | $49.81 \pm 9.63$ |
| | SC(Ours) | $\mathbf{94.41 \pm 0.10}$ | $\mathbf{92.05 \pm 0.48}$ | $75.78 \pm 2.02$ | $\mathbf{87.41 \pm 0.81}$ | $\mathbf{75.97 \pm 1.73}$ | $\mathbf{72.71 \pm 1.20}$ |
| $S_M$ | GCE | $86.51 \pm 0.39$ | $79.74 \pm 0.93$ | $55.83 \pm 0.74$ | $74.03 \pm 0.41$ | $77.98 \pm 0.33$ | $73.10 \pm 0.38$ |
| | SCE | $86.73 \pm 0.54$ | $86.18 \pm 1.21$ | $70.35 \pm 4.03$ | $81.09 \pm 1.20$ | $79.00 \pm 0.72$ | $70.52 \pm 2.67$ |
| | CT+ | $86.32 \pm 0.22$ | $73.35 \pm 0.08$ | $37.50 \pm 0.41$ | $65.72 \pm 0.21$ | $74.97 \pm 0.39$ | $71.34 \pm 0.18$ |
| | ELR | $87.90 \pm 0.74$ | $\mathbf{89.09 \pm 0.84}$ | $71.28 \pm 1.93$ | $82.76 \pm 0.53$ | $77.39 \pm 2.91$ | $72.33 \pm 4.71$ |
| | CDR | $85.87 \pm 1.39$ | $82.84 \pm 1.81$ | $63.75 \pm 4.46$ | $77.48 \pm 1.55$ | $79.16 \pm 1.07$ | $74.21 \pm 1.54$ |
| | QAM | $85.64 \pm 1.26$ | $87.79 \pm 1.15$ | $67.43 \pm 5.73$ | $80.29 \pm 2.49$ | $76.26 \pm 0.72$ | $--$ |
| | CLE | $86.85 \pm 0.84$ | $83.90 \pm 0.93$ | $57.40 \pm 2.52$ | $76.05 \pm 1.04$ | $77.11 \pm 1.55$ | $\mathbf{75.03 \pm 0.77}$ |
| | SC(Ours) | $\mathbf{91.51 \pm 0.62}$ | $87.14 \pm 2.15$ | $\mathbf{72.01 \pm 1.19}$ | $\mathbf{83.55 \pm 0.93}$ | $\mathbf{79.44 \pm 0.65}$ | $72.89 \pm 0.98$ |

Table 2: Mean DSC (in percent) and standard deviation for five models trained on *Cityscapes* datasets and *LIDC-IDRI* dataset. Method with best mean DSC is highlighted.

| Dataset | GCE | SCE | CT+ | ELR | CDR | QAM | CLE | SC(Ours) |
|---|---|---|---|---|---|---|---|---|
| *Cityscapes* | $72.02 \pm 0.14$ | $76.35 \pm 0.13$ | $72.79 \pm 0.14$ | $73.46 \pm 0.17$ | $71.26 \pm 0.16$ | $71.33 \pm 0.14$ | $74.41 \pm 0.12$ | $\mathbf{82.30 \pm 0.10}$ |
| *LIDC-IDRI* | $42.56 \pm 0.58$ | $50.40 \pm 0.15$ | $40.25 \pm 0.11$ | $49.76 \pm 0.40$ | $49.41 \pm 0.79$ | $43.56 \pm 0.29$ | $49.79 \pm 0.19$ | $\mathbf{53.72 \pm 1.70}$ |

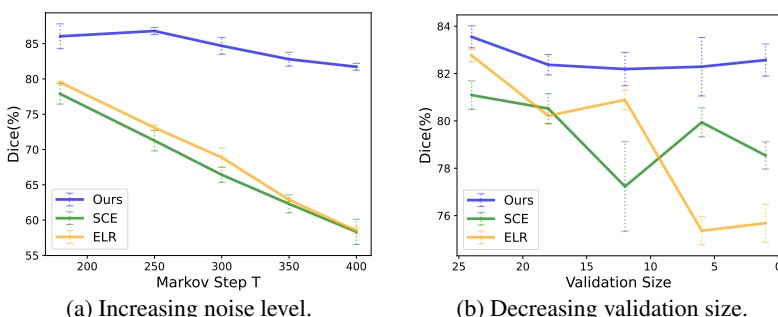

(a) Increasing noise level.      (b) Decreasing validation size.

Figure 6: Ablation study.

**Decreasing Validation Size.** In this experiment, we show that SC works well even with an extremely small clean validation set. Following setting $S_M$ of *JSRT* dataset, we shrink the validation set from 24 to 18, 12, 6, 1, respectively. We compare the performance with SCE and ELR. Results in Figure 6b show that our proposed method still works well when the validation size is extremely small, even if only one clean sample is provided.

CONCLUSION

In this paper, we proposed a Markov process to model segmentation label noise. Targeting such label noise model, we proposed a label correction method to recover true labels progressively. We provide theoretical guarantees of the correctness of a conceptual algorithm and relax it into a more practical algorithm, called **SC**. Our experiments show significant improvements over existing approaches on both synthetic and real-world label noise.

ACKNOWLEDGMENTS

The authors acknowledge the National Cancer Institute and the Foundation for the National Institutes of Health, and their critical role in the creation of the free publicly available LIDC/IDRI Database used in this study.

This research of Jiachen Yao and Chao Chen was partly supported by NSF CCF-2144901. The reported research of Prateek Prasanna was partly supported by NIH 1R21CA258493-01A1. The content is solely the responsibility of the authors and does not necessarily represent the official views of the National Institutes of Health. Mayank Goswami would like to acknowledge support from US National Science Foundation (NSF) grant CCF-1910873.

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

## A    APPENDIX

### A.1    PROOFS

**Lemma 1.** $\phi$ *is the signed distance function defined for domain* $[\mathbb{E}\boldsymbol{Y}]_{\geq 0.5}$. *If* $\theta_3 \ll 0.5$, *then*

$$\tilde{\phi} = \begin{cases} \phi - [\theta_1\theta_2]_{\geq 0.5} - 1, & \text{if } \theta_1 \geq 0.5, \\ \phi + [1 + \theta_1\theta_2 - \theta_2]_{<0.5} + 1 & \text{otherwise.} \end{cases} \tag{A.1}$$

*Here* $\tilde{\phi}$ *is the signed distance function of* $[\mathbb{E}\tilde{\boldsymbol{Y}}]_{\geq 0.5}$ *when* $T = 1$.

*Proof.* When $T = 1$, the given $\boldsymbol{Y}$ is deterministic, we have

$$\mathbb{E}[\tilde{\boldsymbol{Y}}] = \boldsymbol{Y} + \theta_1\theta_2\mathbb{E}[\partial\boldsymbol{B}] + (\theta_1 - 1)\theta_2\mathbb{E}[\partial\boldsymbol{F}] + \theta_3\mathbb{E}[\textbf{Sign}]. \tag{A.2}$$

Recall that $\textbf{Sign} = \boldsymbol{B}^{(T)} \odot \boldsymbol{B} - \boldsymbol{F}^{(T)} \odot \boldsymbol{F}$, indicating $\text{Sign}_s = 0$ if $s \in \partial\boldsymbol{F} \cup \partial\boldsymbol{B}$. Therefore,

$$\mathbb{E}[\tilde{Y}_s] = \begin{cases} \theta_1\theta_2, & s \in \partial\boldsymbol{B} \\ 1 + \theta_1\theta_2 - \theta_2, & s \in \partial\boldsymbol{F} \\ Y_s \pm \theta_3, & \text{otherwise} \end{cases} \tag{A.3}$$

Since $Y_s$ is binary, and if $\theta_s \ll 0.5$, $[Y_s \pm \theta_3]_{\geq 0.5} = Y_s$. Therefore,

$$\tilde{c}(X_s) = \begin{cases} [\theta_1\theta_2]_{\geq 0.5}, & s \in \partial\boldsymbol{B} \\ [1 + \theta_1\theta_2 - \theta_2]_{\geq 0.5}, & s \in \partial\boldsymbol{F} \\ Y_s, & \text{otherwise.} \end{cases} \tag{A.4}$$

We claim the following three facts.

(1) $\theta_1\theta_2 \geq 0.5$ only if $\theta \geq 0.5$.
This is true because if $\theta < 0.5$, $\theta_1\theta_2 < 0.5$ since $\theta_2 \leq 1$.

(2) $1 + \theta_1\theta_2 - \theta_2 < 0.5$ only if $\theta < 0.5$
If $\theta \geq 0.5$, then $1 + (\theta_1 - 1)\theta_2 \geq 1 - 0.5\theta_2 \geq 0.5$.

(3) $\theta_1\theta_2 \geq 0.5$ and $1 + \theta_1\theta_2 - \theta_2 < 0.5$ are mutual exclusive.
First we notice when $\theta_1\theta_2 \geq 0.5$, $1 + \theta_1\theta_2 - \theta_2 \geq 1.5 - \theta_2 \geq 0.5$. Then when $1 + \theta_1\theta_2 - \theta_2 < 0.5$, $\theta_1\theta_2 < \theta_2 - 0.5 < 0.5$.

The first two facts separate the two cases in Equation A.1. And according to fact (3), either $\partial\boldsymbol{B}$ is flipped into foreground or $\partial\boldsymbol{F}$ is flipped into background. In the former case, expansion happens because $\theta_1 > 0.5$, and $\partial\boldsymbol{B}$ becomes $\partial\tilde{\boldsymbol{F}}$, leading to $\tilde{\phi} = \phi - [\theta_1\theta_2]_{\geq 0.5} - 1$. Opposite happens in the latter case. Therefore,

$$\tilde{\phi} = \begin{cases} \phi - [\theta_1\theta_2]_{\geq 0.5} - 1, & \text{if } \theta_1 \geq 0.5, \\ \phi + [1 + \theta_1\theta_2 - \theta_2]_{<0.5} + 1 & \text{otherwise.} \end{cases} \tag{A.5}$$

Proof done. $\qquad\square$

**Theorem 1.** *If* $\exists \varepsilon_0, \varepsilon_1 > 0$, *s.t.*

$$\mathbb{E}_{\boldsymbol{X}}\left[\sup_s |\hat{\phi}(X_s) - \tilde{\phi}(X_s)|\right] \leq \varepsilon_0, \tag{A.6}$$

*and*

$$\sup_{\boldsymbol{X}}\left[\sup_s |\hat{\phi}(X_s) - \tilde{\phi}(X_s)|\right] \leq \varepsilon_1, \tag{A.7}$$

*hold for the learned classifier* $\hat{c}(\boldsymbol{X})$, *then* $\forall \varepsilon > \varepsilon_0$ *and for a fixed confidence level* $0 \leq \alpha \leq 1$, *with*

$$V \geq \frac{\varepsilon_1^2}{2(\varepsilon - \varepsilon_0)^2} \log\left(\frac{2|I|}{\alpha}\right) \tag{A.8}$$

*number of clean samples* $\{\mathbf{x}_v, \mathbf{y}_v\}_{v=1}^V$, $\phi'$ *can be recovered within* $\varepsilon + \varepsilon_0$ *error with probability at least* $1 - \alpha$, *i.e.* $P(\mathbb{E}_{\boldsymbol{X}}[\sup_s |\phi'(X_s) - \phi(X_s)|] \leq \varepsilon + \varepsilon_0) \geq 1 - \alpha$.

*Proof.* To bound the label correction error, we first aim to bound the bias estimation error. We try to prove that

$$P(|\hat{\Delta} - \Delta| \geq \varepsilon) \leq \alpha. \tag{A.9}$$

Recall that the definition of $\hat{\Delta}$ is

$$\hat{\Delta} = \frac{1}{V} \sum_{v=1}^{V} \frac{1}{|I|} \sum_{s \in I} (\hat{\phi}_s^v - \phi_s^v), \tag{A.10}$$

and $\Delta$ is the true bias that is the same for every image $\boldsymbol{X}$. Hence,

$$\Delta = \frac{1}{V} \sum_{v=1}^{V} \frac{1}{|I|} \sum_{s \in I} (\tilde{\phi}_s^v - \phi_s^v). \tag{A.11}$$

Take Equation A.10 and Equation A.11 into the LHS of Inequality A.9,

$$
\begin{aligned}
P(|\hat{\Delta} - \Delta| \geq \varepsilon) &= P\left( \left| \frac{1}{V} \sum_v \frac{1}{|I|} \sum_s (\hat{\phi}_s^v - \tilde{\phi}_s^v) \right| \geq \varepsilon \right) \\
&\leq P\left( \frac{1}{V} \sum_v \frac{1}{|I|} \sum_s |\hat{\phi}_s^v - \tilde{\phi}_s^v| \geq \varepsilon \right).
\end{aligned}
\tag{A.12}
$$

Therefore, to prove A.9 is equivalent to proving

$$P\left( \frac{1}{V} \sum_v \frac{1}{|I|} \sum_s |\hat{\phi}_s^v - \tilde{\phi}_s^v| \geq \varepsilon \right) \leq \alpha. \tag{A.13}$$

Given an index $s$, the absolute error $|\hat{\phi}_s^v - \tilde{\phi}_s^v|$ is *i.i.d.* over different images $\boldsymbol{X}_v$. And the error is bounded by the image size, i.e. $0 \leq |\hat{\phi}_s^v - \tilde{\phi}_s^v| \leq \varepsilon_1$. According to *Hoeffding's inequality*, a lower bound is provided for the following probability,

$$P\left( \left| \frac{1}{V} \sum_v |\hat{\phi}_s^v - \tilde{\phi}_s^v| - \mathbb{E}_{\boldsymbol{X}}\left[ |\hat{\phi}_s^v - \tilde{\phi}_s^v| \right] \right| \geq \varepsilon \right) \leq 2\exp\left[ -\frac{2V\varepsilon^2}{\varepsilon_1^2} \right]. \tag{A.14}$$

By the given model error A.6, $\forall s \in I$,

$$\mathbb{E}_{\boldsymbol{X}}\left[ |\hat{\phi}_s^v - \tilde{\phi}_s^v| \right] \leq \mathbb{E}_{\boldsymbol{X}}\left[ \sup_s |\hat{\phi}_s^v - \tilde{\phi}_s^v| \right] \leq \varepsilon_0. \tag{A.15}$$

Combine Inequality A.14 and A.15,

$$P\left( \frac{1}{V} \sum_v |\hat{\phi}_s^v - \tilde{\phi}_s^v| \geq \varepsilon \right) \leq 2\exp\left[ -\frac{2V(\varepsilon - \varepsilon_0)^2}{\varepsilon_1^2} \right]. \tag{A.16}$$

Observing the LHS of A.13 and A.16, the difference is that random variable in A.13 is the average error among a specific group of indices, while A.16 holds for arbitrary index. If we iterate A.16 over the index set, then for a group of indices, A.13 naturally holds. In other words,

$$\left( \frac{1}{V} \sum_v \frac{1}{|I|} \sum_s |\hat{\phi}_s^v - \tilde{\phi}_s^v| \geq \varepsilon \right) \subseteq \bigcup_{s \in I} \left( \frac{1}{V} \sum_v |\hat{\phi}_s^v - \tilde{\phi}_s^v| \geq \varepsilon \right). \tag{A.17}$$

Hence,

$$
\begin{aligned}
P\left( \frac{1}{V} \sum_v \frac{1}{|I|} \sum_s |\hat{\phi}_s^v - \tilde{\phi}_s^v| \geq \varepsilon \right) &\leq \sum_s P\left( \frac{1}{V} \sum_v |\hat{\phi}_s^v - \tilde{\phi}_s^v| \geq \varepsilon \right) \\
&\leq 2|I|\exp\left[ -\frac{2V(\varepsilon - \varepsilon_0)^2}{\varepsilon_1^2} \right],
\end{aligned}
\tag{A.18}
$$

To obtain A.13, let

$$2|I|\exp\left[ -\frac{2V(\varepsilon - \varepsilon_0)^2}{\varepsilon_1^2} \right] \leq \alpha, \tag{A.19}$$

and if

$$V \geq \frac{\varepsilon_1^2}{(\varepsilon - \varepsilon_0)} \log \frac{2|I|}{\alpha}, \tag{A.20}$$

then

$$P(|\hat{\Delta} - \Delta| \geq \varepsilon) \leq \alpha. \tag{A.21}$$

Next we are about bounding the label correction error. Note that $\phi = \tilde{\phi} - \Delta$, and our label correction step indicates that $\phi' = \hat{\phi} - \hat{\Delta}$. Then we have,

$$
\begin{aligned}
\mathbb{E}_{\boldsymbol{X}} \left[ \sup_s |\phi'(X_s) - \phi(X_s)| \right] &= \mathbb{E}_{\boldsymbol{X}} \left[ \sup_s |(\hat{\phi}(X_s) - \tilde{\phi}(X_s)) + (\Delta - \hat{\Delta})| \right] \\
&\leq \mathbb{E}_{\boldsymbol{X}} \left[ \sup_s |(\hat{\phi}(X_s) - \tilde{\phi}(X_s))| \right] + |\Delta - \hat{\Delta}| \\
&\leq \varepsilon_0 + |\Delta - \hat{\Delta}|.
\end{aligned}
\tag{A.22}
$$

Therefore,

$$
\begin{aligned}
P(|\hat{\Delta} - \Delta| \geq \varepsilon) &= P(|\hat{\Delta} - \Delta| + \varepsilon_0 \geq \varepsilon + \varepsilon_0) \\
&\geq P \left( \mathbb{E}_{\boldsymbol{X}} \left[ \sup_s |\phi'(X_s) - \phi(X_s)| \right] \geq \varepsilon + \varepsilon_0 \right).
\end{aligned}
\tag{A.23}
$$

According to A.21, we proved

$$P \left( \mathbb{E}_{\boldsymbol{X}} \left[ \sup_s |\phi'(X_s) - \phi(X_s)| \right] \geq \varepsilon + \varepsilon_0 \right) \leq \alpha. \tag{A.24}$$

$\square$

## A.2 Implementation Details

All the experiments are done on one NVIDIA GTX 1080 GPU (12G Memory) with a batch size of 2. For JSRT, we use a learning rate of $0.05$. At 750 iter, we decrease the learning rate by a factor of 10, with total 1.5K iterations. For ISIC 2017 and Cityscapes, the learning rate is also set to be $0.05$, and decay is done every 4K iterations for ISIC 2017 and 3.1K iterations for Cityscapes, by a factor of 2. The model converges at around 10K iterations.

### A.2.1 Noise and Dataset Settings.

**Synthetic Noise Setting.** The synthetic noise follows parameters in Table A.1. $S_E$ stands for expansion setting and $S_S$ standing for shrinkage setting. And for the ablation study of noise level, we keep the same $\theta_1, \theta_2$ in $S_E$ of JSRT dataset, and increase $T$ from $(180, 180, 100)$ (for lung, heart, clavicle) to $(250, 250, 170)$, $(300, 300, 220)$, $(350, 350, 270)$, $(400, 400, 320)$, respectively. $\theta_3$ is set $0.1$ for all settings. To improve the smoothness, we also use a Gaussian filter before the random flipping. An example for increasing noise level of heart class is shown in Fig. A.1.

Table A.1: Label noise settings on two synthetic datasets. $M(T, \theta_1, \theta_2)$ stands for the proposed multi-step Markov process.

| Noise | JSRT | | | ISIC | Brats |
|---|---|---|---|---|---|
| Setting | Lung | Heart | Clavicle | 2017 | 2020 |
| $S_E$ | $M(180, 0.7, 0.03)$ | $M(180, 0.7, 0.03)$ | $M(100, 0.7, 0.03)$ | $M(200, 0.8, 0.05)$ | $M(80, 0.7, 0.05)$ |
| $S_S$ | $M(200, 0.3, 0.05)$ | $M(200, 0.3, 0.05)$ | $M(120, 0.3, 0.05)$ | $M(200, 0.2, 0.05)$ | $M(80, 0.3, 0.05)$ |

**Dataset Settings.** For *Brats 2020* dataset, we merge all three classes into a single class. The reason is that some classes are so few in volumes that most of them could vanish when we create the synthetic noisy labels. For *LIDC-IDRI* dataset, we follow the pre-processing of Kohl et al. (2018) by extracting 2D $128 \times 128$ slices centred around the annotated nodules, resulting in 8843 images in the training set, 1993 images in the validation set and 1980 images in the test set. We set $\gamma = 1$ in our algorithm and it terminates after 1 iteration. For *Citysacpes* dataset, since the test labels are not publicly available, we use the validation set as test set. We resize all images to $256 \times 512$ and remove

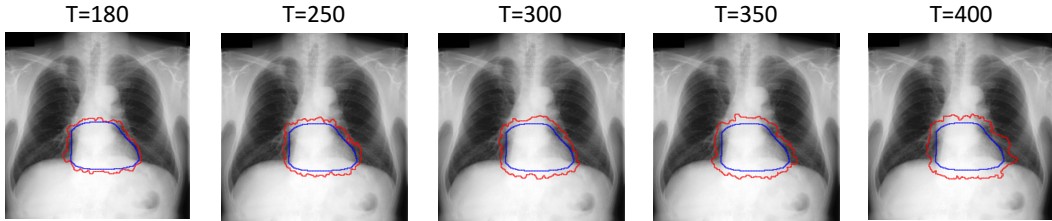

Figure A.1: The increasing noise level in 'heart' class. Red line is noisy boundary and blue line is true boundary.

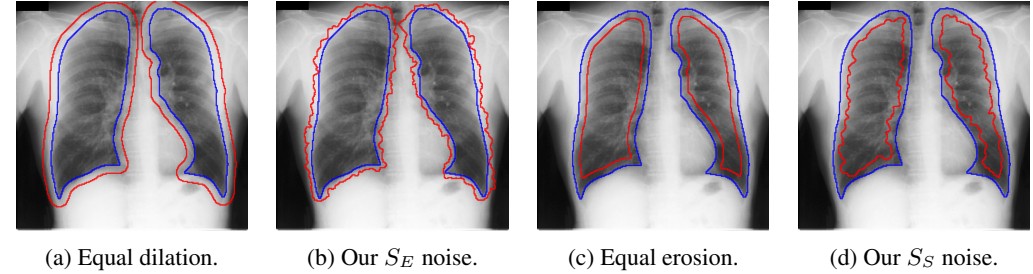

(a) Equal dilation.     (b) Our $S_E$ noise.     (c) Equal erosion.     (d) Our $S_S$ noise.

Figure A.2: Compare equal dilation (a) and equal erosion (c) with Markov noise (b) and (d) generated by our Markov model.

images whose coarse masks have few or no car labels. This gives us 2086 training images and 350 test images. We further randomly split the training images into 1986 training images (with noisy label) and 100 validation images (with clean label). We also use a simple U-Net as our backbone network. We conservatively choose a small $\gamma = 0.4$, and correct label noise progressively. The algorithm terminates after 2 iterations.

### A.2.2 BASELINES.

We compare the proposed SC with current SOTA methods from both classification context and segmentation context: (1)GCE Zhang & Sabuncu (2018) trains the deep neural networks with a generalized cross entropy loss to handle noisy labels. The hyperparameters $k, q$ in this work are set to be $0.5$ and $0.8$, respectively. (2)SCE Wang et al. (2019b) combines the cross entropy and reverse cross entropy (RCE) into a single noise robust loss. The hyperparameters $\alpha$ and $\beta$ are set to be $1.0$ and $0.5$ for JSRT and ISIC 2017 dataset, and $\alpha = 1.0$, $\beta = 0.5$ for Cityscapes dataset. (3)CT+ Yu et al. (2019) utilizes two networks and selects small-loss instances for cross training. To employ this method into segmentation, we treat each pixel as an instance. The prior estimated noise rate $\tau$ is estimated with the clean validation set. (4)ELR Liu et al. (2020) utilizes the early stopping technique into a regularization term to prevent the network from memorizing noisy labels. The hyperparameters $\lambda$ and $\beta$ are set to be 7 and $0.8$, respectively. (5)QAM Zhu et al. (2019) re-weights samples with a quality awareness module (QAM), trained together with the segmentation model. The outputs of QAM are the weights for each image in the loss function. (6)CLE Zhang et al. (2020b) leverages confident learning to correct noisy labels based on the network prediction. Then train a robust network with corrected labels.

### A.3 ILLUSTRATIVE EXPERIMENTS

**Network predictions can be over confident if bias shows up in training labels.** Some works (Zhang et al., 2020b; Li et al., 2021) choose to trust the network prediction probability to correct label noises, i.e. they believe predictions with small confidence is likely to be wrong while pixels with large confidence tend to be correctly predicted. However, the network can fit to noisy labels quickly and be overconfident when trained with biased noisy labels. This phenomenon is also observed by Zhang et al. (2016). In Figure A.3(a), we train a network with dilated noises and show its prediction probability map, i.e. the output after *sigmoid*. The red pixel is predicted as foreground

with a high probability $0.9994874$, whereas it is actually in background. Therefore, methods based on trusting the network predictions cannot correct this label because the network is over confident. And since the network can fit to noise rapidly, early learning techniques also cannot eliminate this bias. Our method works because we do not trust the network prediction. Instead, we compare it to the clean label in validation set and estimate the bias. We then eliminate this bias in the training prediction. We also show how the prediction probability changes while training the model in Figure A.3(b). It shows that the model can be over confident quickly while training. So methods that employ early learning techniques (Liu et al., 2020; Arpit et al., 2017; Liu et al., 2022) are hard to work under biased noisy labels.

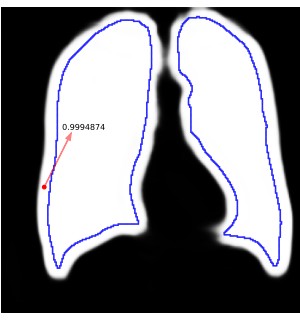

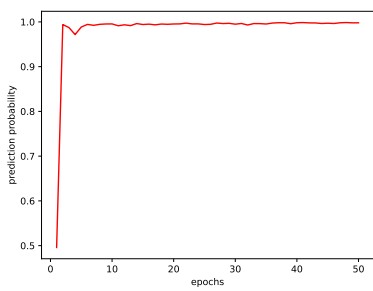

(a) Prediction probability.

(b) The probability of the red point while training.

Figure A.3: A model trained with biased labels can be over confident. The model is trained with dilated noises. In the left figure, the blue line is the true segmentation boundary, and the red point is the selected pixel where it is supposed to be background (in the true label) but is predicted as foreground. The right figure shows how the network prediction on the red point fits to noise along training.

**Noisy annotations with random inner holes.** Our Markov noise model assumes the spatial bias occurs around the true boundary, but it does not reject other random noises like inner holes. Figure A.4 shows an expanded noise with low-frequency inner holes (1st row), and our method can correct this hole after 2 iterations.

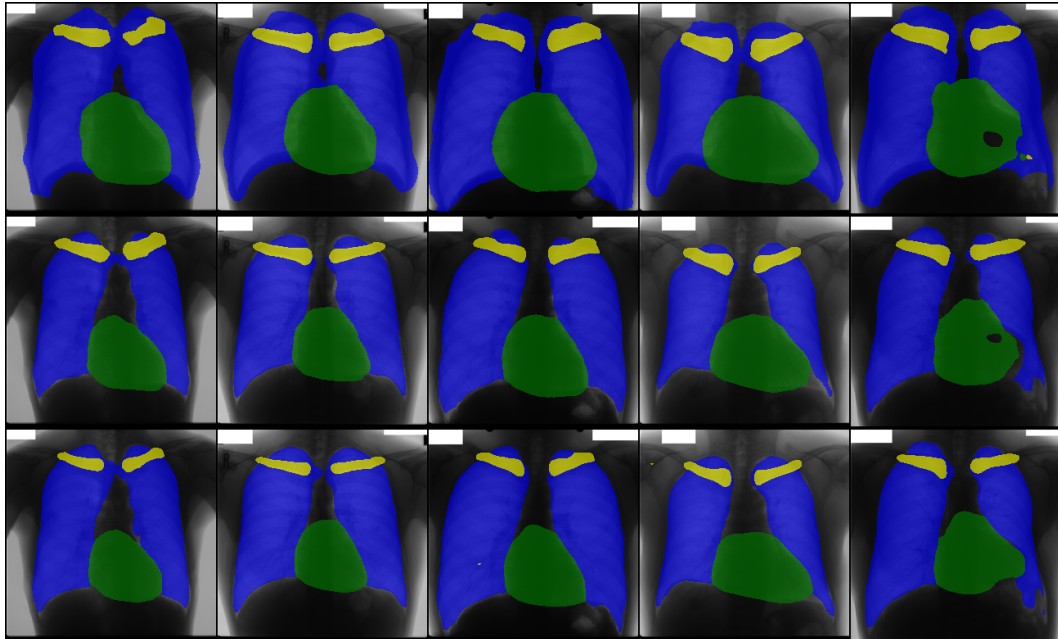

Figure A.4: A model trained with noisy labels with random inner holes (first row). The corrected label by the proposed method after 1 iteration (second row) and 2 iterations (third row).

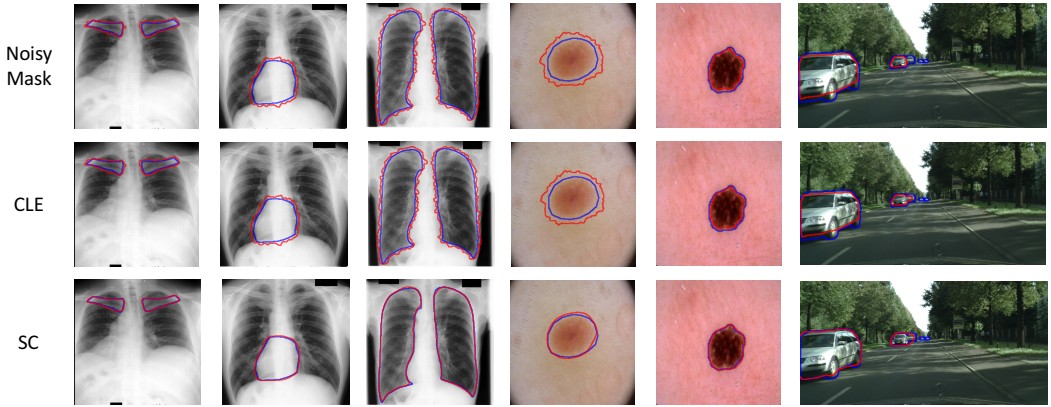

Figure A.5: Label correction results from the baseline CLE and our method SC. Red lines are corrected boundaries, and blue lines are true boundaries.

## A.4 QUALITATIVE RESULTS

We provide qualitative results for prediction on test images in Fig. A.6 and Fig. A.7 as two sub parts. We also show the label correction on training images in Fig. A.5.

## A.5 EXTENTED NOTATION

Although our notation is for 2D images, our method naturally generalizes to 3D. For 3D input $X \in \mathbb{R}^{H \times W \times D \times C}$, to define the spatial correlation in 3D input, we just need to redefine the neighbor $N_s$ as the six-neighbor, i.e. the four-neighbor in the same slice and up and down pixel in adjacent slices. In general, our spatial correlation can be defined for images of arbitrary dimension.

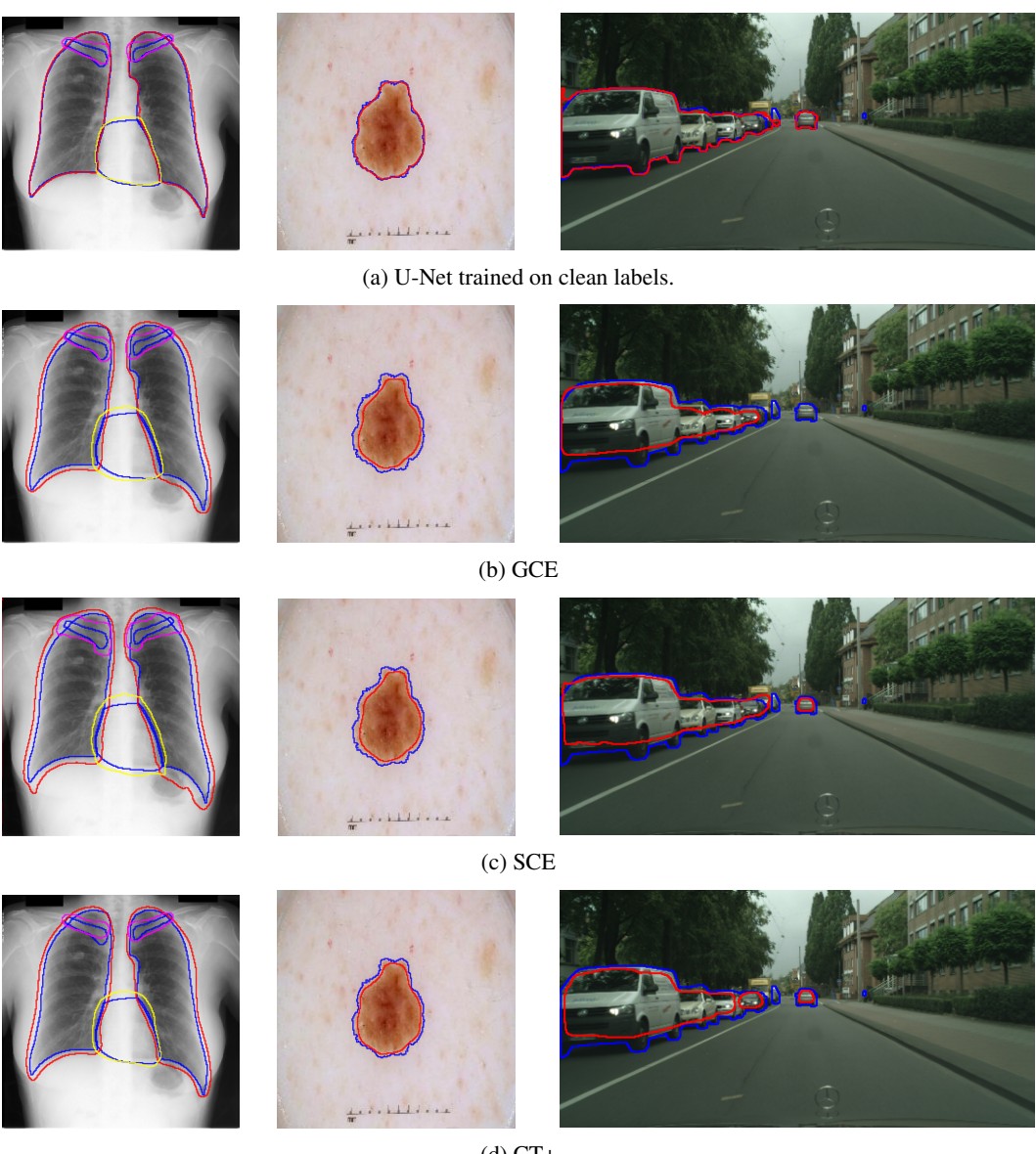

(a) U-Net trained on clean labels.

(b) GCE

(c) SCE

(d) CT+

Figure A.6: Qualitative results of different model predictions – part one. Blue lines are true boundaries. All other colors are corresponding prediction boundaries.

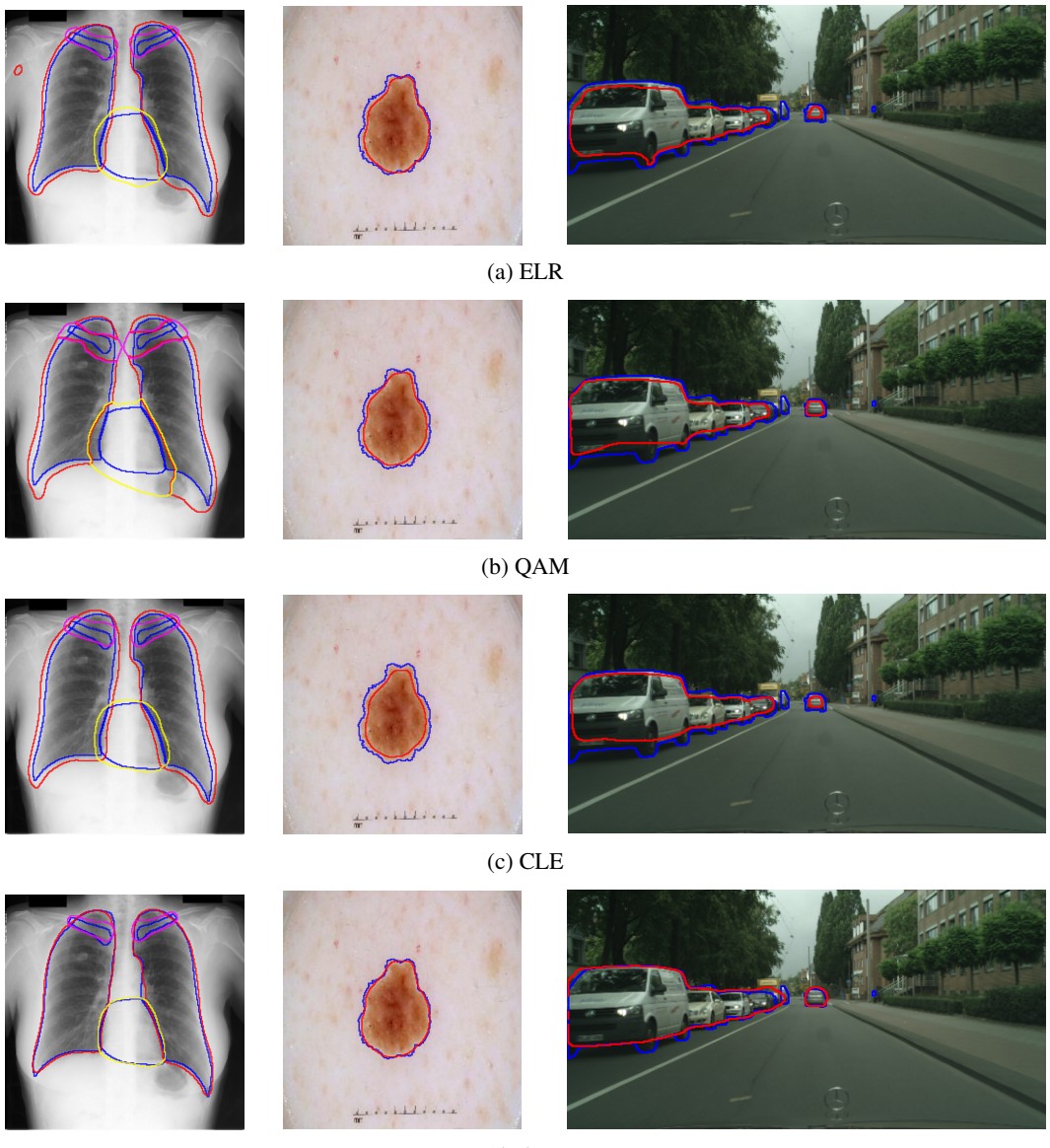

Figure A.7: Qualitative results of different model predictions – part two. Blue lines are true boundaries. All other colors are corresponding prediction boundaries.

