# OpenReview forum: "Learning to Segment from Noisy Annotations: A Spatial Correction Approach"
_ICLR.cc/2023/Conference — ICLR 2023 poster_

### Official Review · Reviewer_VNP8 · 2022-10-21

**Confidence:** 5
**Correctness:** 2
**Technical Novelty And Significance:** 2
**Empirical Novelty And Significance:** 2
**Recommendation:** 6

**Clarity, Quality, Novelty And Reproducibility:**

The contributions are only marginally significant or novel.


**Strength And Weaknesses:**

Strength:
The Markov model for segmentation label noise considers spatial correlation.

Weakness:
The spatial correlation proposed by the authors contradicts the input 2D image. Whether the authors consider three-dimensional spatial correlations.


**Summary Of The Paper:**

This work proposed a label correction method to tackle the label noise in the segmentation task.  They propose an algorithm to correct the Markov label noise.


**Summary Of The Review:**

This paper proposed a Markov process to model segmentation label noise. The issue that this paper focuses on are important but not very attractive. The method needs to be extended for better tackle the problem of the 2D/3D segmentation label noise. Whether it is more convincing to experiment with real data labeled by different intern doctors. More specifically,
1. It is strongly recommended that the input be changed to 3D voxel segmentation to match SC.
2. The results of the Brats 2020 dataset should show multi-class results.  Is this method capable of handling the multi-class segmentation label noise problem?
3. Fig.3 is hard to follow with the 3-step Markov process.

---

> ### Author Response · Authors · 2022-11-19
> **Response to Reviewer VNP8**
>
> Thank you for the careful reading of our paper and detailed feedback. We hope we can address your concerns with additional experiments and following discussions.
>
> Q1. **The spatial correlation proposed by the authors contradicts the input 2D image. Whether the authors consider three-dimensional spatial correlations.**
> * Thank you for this question. We added clarification in Section A.4 of Appendix (Page 18). Our spatial correlation in Section 3 is defined in 2D. If we understand correctly, your concern might come from the third dimension $C$ in the notation $\mathbf{X}\in\mathbb{R}^{H\times W\times C}$. $C$ represents the number of channels. For example $C=3$ (RGB) in the ISIC dataset and $C=1$ in the JSRT dataset. Both of them are still 2D images. The spatial correlation is defined between a pixel $s$ and its neighbor $N_s$. In 2D images, $N_s$ is defined as the four-neighbor, i.e. left, right, top and bottom. In an RGB image, the four-neighbor is only defined in the spatial dimension, i.e. in $\mathbb{R}^{H\times W}$. The RGB channels are considered as a single pixel with a vector value.
> * Although our notation is for 2D images, our method naturally generalizes to 3D. For 3D input $X\in\mathbb{R}^{H\times W\times D\times C}$, it has height, width and depth three spatial dimensions and a channel dimension. For example, Brats 2020 dataset is a 3D segmentation set with $C=4$. To define the spatial correlation in 3D input, we just need to redefine the neighbor $N_s$ as the six-neighbor, i.e. the four-neighbor in the same slice and up and down pixel in adjacent slices. In general, our spatial correlation can be defined for images of arbitrary dimension.
> * In the experiment, we apply the 2D version of our method for 2D inputs like JSRT, ISIC and Cityscapes. We also apply the 3D version of our method to 3D inputs like Brats 2020. We generate noisy labels for both 2D and 3D images. Our results also show the proposed method outperforms other baselines in both 2D and 3D images.
>
> Q2. **The method needs to be extended to better tackle the problem of the 2D/3D segmentation label noise. It is strongly recommended that the input be changed to 3D voxel segmentation to match SC.**
> * We hope our discussion in Q1 can address your concerns about the input dimension.
>
> Q3. **It is more convincing to experiment with real data labeled by different intern doctors.**
> * Thank you for this suggestion. We added a new experiment on the LIDC-IDRI dataset to address your concern. Please refer to *Q2 in the summary response*.
>
> Q4. **The results of the Brats 2020 dataset should show multi-class results. Is this method capable of handling the multi-class segmentation label noise problem?**
> * Thank you for pointing this out. The proposed method is capable of handling the multi-class segmentation label noise problem. Our JSRT dataset is a 3-class segmentation dataset. We show in Table 1 (Page 9) that our method outperforms baselines in all settings w.r.t. the total Dice Score. As for Brats 2020, we merge all foreground classes into a single class. This is because masks of certain classes are too small to have meaningful erosion/dilation noise. We have added the explanation in Appendix A.2.1 (Page 16).
>
> Q5. **Fig.3 is hard to follow with the 3-step Markov process.**
> * We provided an explanation for Fig.3 in Section 3 first paragraph (Page 3). We provide more clarifications here.
> * Fig.3 has four subfigures (a)-(d). (a) is the original (true) label mask, where red pixels are foreground and blue pixels are background. We mark the background and foreground boundary pixels as $\partial B$ and $\partial F$, respectively. (b) is the first step of the Markov process. It is an expansion step, so pixels marked as '1' have been flipped into the foreground. The flipped pixels are randomly chosen with probability $\theta_2$ from the $\partial B$ pixels in (a). The second step in (c) is also an expansion step, where pixels marked as '2' are flipped into foreground. The flipped pixels are randomly chosen from the background boundary of (b). The third step (d) is a shrinkage step. Pixels marked as '3' were foreground in (c) but are flipped into background in (d). These pixels are randomly chosen from the foreground boundary of (c).
> * We have added these details to the caption of Fig.3 to improve readability.

---

> ### Author Response · Authors · 2022-12-06
> **Thank you for your positive feedback**
>
> We appreciate your support on our work. To address your concerns, we provided more experiments and explanations. More specifically, we
>
> 1) clarified why our method can be applied to both 2D and 3D
>
> 2) added another real-world experiment on LIDC-IDRI dataset
>
> If you have any concern, please kindly let us know. We are looking forward to your further response.

---

> > ### Comment · Reviewer_VNP8 · 2022-12-07
> > **My concerns have been addressed.**
> >
> > Overall, it looks pretty good now. My concerns have been addressed.

---

> > > ### Author Response · Authors · 2022-12-08
> > > **Thank you very much!**
> > >
> > > Thank you very much for your support!
> > >
> > > Paper 3111 Authors

---

### Official Review · Reviewer_VpVv · 2022-10-24

**Confidence:** 4
**Correctness:** 4
**Technical Novelty And Significance:** 4
**Empirical Novelty And Significance:** 4
**Recommendation:** 6

**Clarity, Quality, Novelty And Reproducibility:**

On clarity, I find the work to be relatively easy to read and follow.

On quality, the limited experimentation on real data makes me question how broadly these results generalize.

On novelty, the approach claims to be the first noise model that is tailored for segmentation tasks. Have no evidence otherwise.

On reproducbility, it is a shame the authors have not chosen to share code to reproduce the experiments.


**Strength And Weaknesses:**

Strengths
------------
- The approach addresses the important and relevant problem of learning to segment from noisy labels.

- The proposed method seems relatively simple and the ideas underlying it makes sense.

- The approach is compared to a wide range of relatively recent prior work and shows superior results.

Weaknesses
----------
- Sentences in abstract seem condenced to the point where some meaning is lost.

- Table 1 list Dice scores of the model along with other methods for learning with noisy labels in datasets with artificial noise added. Noise is added according to the noise models that is also used by the proposed approach to remove the noisy labels. While this is of course a nice sanity check of the proposed approach, it is perhaps not surprising that the proposed approach would excel at this. I would suggest the authors discuss the limitations of this experiment more clearly.

- Only a single experiment is done with real noisy labels. I would have appreciated more experimentation on real data. Noisy labels in segmentation is generally the rule rather than the exception, particularly with medical images. I think i
t would have greatly strengthened the manuscript if the authors had made a greater effort to show their method can help to solve this problem.

- Code is not provided, which could hurt reproducibility.


**Summary Of The Paper:**

The paper proposed a method for learning image segmentation from noisy annotation. The noise model proposed models human annotation errors that are spatially correlated and dependent on local image features. Results of the approach is compared to a wide range of prior work using artificially added noise in three different datasets and on a single dataset with real noise.

**Summary Of The Review:**

I have put the paper marginally below the acceptance threshold. I do like the paper and the approach, but I would have expected a little bit more in terms of showing that it works with real data.

---

> ### Author Response · Authors · 2022-11-19
> **Response to Reviewer VpVv**
>
> Thank you very much for the constructive feedback. We appreciate that you like our work and find the problem we are addressing is important. We have run additional experiments and enhanced the paper according to your suggestions. Below we hope to address your concerns one-by-one.
>
> Q1. **Sentences in abstract seem condensed to the point where some meaning is lost.**
> * Thank you for pointing this out. We have improved the abstract accordingly.
>
> Q2. **The synthetic experiments only use label noise generated by the proposed noise model. It is perhaps not surprising that the proposed approach would excel at this.**
> * Some of our synthetic noise settings are following previous SOTA methods. Please refer to *Q1 in the summary response*.
>
> Q3. **Only a single experiment is done with real noisy labels.  I would have appreciated more experimentation on real data. Noisy labels in segmentation is generally the rule rather than the exception, particularly with medical images.**
> * We added a new experiment on the LIDC-IDRI dataset to address your concern. Please refer to *Q2 in the summary response*.
>
> Q4. **Code is not provided, which could hurt reproducibility.**
> * We will release our code upon acceptance.

---

> ### Author Response · Authors · 2022-12-06
> **Thank you for your positive feedback**
>
> Thank you for your support on our work. We made the following improvements based on your initial review.
>
> 1) We did additional experiments on another real-world dataset.
>
> 2) We clarified our synthetic settings.
>
> 3) We improved our abstract in the manuscript.
>
> Please kindly let us know if you have further concerns.

---

### Official Review · Reviewer_LKDX · 2022-10-27

**Confidence:** 3
**Correctness:** 2
**Technical Novelty And Significance:** 3
**Empirical Novelty And Significance:** 3
**Recommendation:** 6

**Clarity, Quality, Novelty And Reproducibility:**

The paper is generally easy to follow. The clarity could be improved at places. I believe the method to be novel and most details are captured to ensure reproducibility.

**Strength And Weaknesses:**

Strengths:
- The paper is well-written and easy to follow.
- The method appears to be novel and produces competitive or superior performance on the shown experiments.
- The method is an interesting step towards spatially-aware modelling of segmentation noise.

Weaknesses:
- The experiments seem like they might be biased. The medical imaging experiments only use the proposed noise model to add synthetic label noise. How would the model perform with a mismatch in label noise? One potential avenue for studying this would be to use a dataset such as LIDC-IDRI that comes with multiple annotators and define the ground truth as a majority voting of the individual labels and select a single rater as the noisy annotations.
- The method implicitly assumes contiguous regions without holes to be segmented. How would this method perform with potential holes within the segmentation masks?


Misc:
- In the related work you mention that superpixel based approaches are not capable of addressing the challenges of noisy segmentation labels. Could you elaborate on this?
- I am curious how this method would compare to weak segmentation methods that only use bounding boxes etc as labels. One could argue that the problems that are solved in both cases are very similar.
- I am curious about the relationship between this method and methods that aim to model spatially aware uncertainty of the segmentation predictions (e.g. [1]). Would spatial awareness in the modelling be sufficient to then calibrate the predictions based on the estimated bias of the annotations?
- p3: "net-work learning" -> "network learning"; "Section 3.1, We aim" -> "we aim"; the penultimate paragraph could use a bit of editing
- p4: "N_S" is only introduced in definition 2 but used earlier
- Section 3.1 could be improved by adding a bit more intuition about the equations.
- Notation: I generally would prefer reading $\mathbb{E}[Y]$ than $\mathbb{E}Y$
- p5: exits -> exists ?
- page 8 mentions sparse labels - what do you mean by that?
- How many seeds do you use for the standard deviations?

[1] Monteiro, Miguel, et al. "Stochastic segmentation networks: Modelling spatially correlated aleatoric uncertainty." Advances in Neural Information Processing Systems 33 (2020): 12756-12767.

**Summary Of The Paper:**

The paper presents an approach to correct noise annotation for segmentations by making use of spatially-aware operations about flipping labels on the boundary of the segmentation. The proposed method uses clean validation data to calculate the bias of the predicted segmentations and uses the specific noise model together with the inferred noise parameters to correct the predictions of the trained segmentation model. This process can be applied iteratively until the predictions are unbiased. The paper compares the proposed method on three medical imaging datasets (ISIC, JSRT and BRATS) as well as the cityscapes dataset and shows superior performance to relevant baselines. The experiments on the medical imaging datasets introduce synthetic noise to the segmentations while the cityscapes task uses coarse-grained segmentations to predict the fine-grained ones.

**Summary Of The Review:**

The paper introduces a novel method for training with noisy segmentation labels but might be biased in its evaluation and comparison to relevant baseline methods. Furthermore, I am not sure whether the assumptions made for the noise-model are as widely applicable as claimed.

---

> ### Author Response · Authors · 2022-11-19
> **Response to Reviewer LKDX (1/2)**
>
> Thank you for your detailed evaluation and constructive suggestions. We also appreciate that you find our work interesting and novel. We hope to address your concerns with additional experiments and following discussions.
>
> Q1. **The experiments seem like they might be biased. The medical imaging experiments only use the proposed noise model to add synthetic label noise. How would the model perform with a mismatch in label noise?**
> * We would like to point out that some of our synthetic noise settings are following previous SOTA methods. Please refer to *Q1 in the summary response*.
> * We appreciate your suggestion on turning LIDC-IDRI into a label noise setting. We add a new experiment creating noisy labels using combinations of human annotations. Please refer to *Q2 in the summary response*.
>
> Q2. **The method implicitly assumes contiguous regions without holes to be segmented. How would this method perform with potential holes within the segmentation masks?**
> * Thank you for the thoughtful question. Aside from spatial perturbation around the boundary, our method does admit other random flipping of pixels which creates inner holes within segmentation masks. The random flipping term $\mathbf{\epsilon}$ in Definition 1 (Page 4) is designed for such cases. Theoretically, when these random noises are in low-frequency, they will not affect the Bayes optimal so that we can still trust the network predictions.
> * We add a result as in Figure A.4 of Appendix A.3 (Page 18). We train a network with expanded noise from our noise model and low-frequency inner holes, and show an example that our method can correct the inner holes after 2 iterations.
>
> Q3. **Why is the superpixel based approach (Li et al., 2021) not capable of addressing the challenges of noisy segmentation labels?**
> * Although the superpixel based approach encodes spatial correlation into their method, they are still not able to solve the bias problem. Their method employs the network prediction at the superpixel level to correct label noise. But chances are that the whole superpixel is predicted wrongly and still with a high confidence.
> * We illustrate this in Figure A.3 of Appendix A.3 (Page 18) where we train a model with dilated labels. The red pixel in Figure A.3(a) is supposed to be background but is predicted as foreground with a high probability (0.9994874). Methods that trust the network predictions cannot flip this label because the network is too confident of its prediction.
>
> Q4. **How is the proposed method compared to weakly supervised segmentation methods that use bounding boxes etc as labels. One could argue that the problems that are solved in both cases are very similar.**
> * This question is quite inspiring. Our method and weakly supervised segmentation methods do share similarities. These methods all train a segmentation model with inaccurate annotations. However, our method tackles this problem by explicitly modeling the noisy label distribution, whereas weakly supervised segmentation methods solve the problem with weaker noise assumptions.
> * On the one hand, weak segmentation labels like bounding boxes can be considered as an expanded noise with both spatial correlation and bias. Our method is capable of solving such label noise to some extent. However, when the box is too large and the noise model assumption does not hold, the network prediction could be completely corrupted. Our method will degrade.
> * On the other hand, weakly supervised segmentation methods can potentially solve the label noise problem. However, they are not leveraging strong noise assumptions and thus may not beat label noise methods.
>
> Q5. **What is the relationship between this method and methods (e.g. [1]) that aim to model spatially aware uncertainty of the segmentation predictions?**
> * Thank you for this insightful question. We believe the difference/connection between our work and the spatially aware uncertainty methods is analogy to the difference between label dependent and data dependent label noise setting. Our spatial correlation lies between noisy label $\tilde{\mathbf{y}}$ and the true label $\mathbf{y}$, i.e. in the probability $P(\tilde{\mathbf{y}}|\mathbf{y})$, which is label-dependent. The spatially aware uncertainty methods directly model the spatial correlation in $P(\mathbf{y}|\mathbf{x})$ that is data-dependent. While the data-dependent spatial correlation is more general, it also requires a large dataset to train. Our label-dependent assumption is much simpler, yet sufficient to capture the essence of real world segmentation label noise. The simplicity of the model allows us to estimate the bias with even only a single validation image, while still achieving superior empirical performance over baselines. However, if apply a data-dependent model to estimate a data-dependent bias, the validation size required will be proportion of the model complexity.

---

> > ### Author Response · Authors · 2022-11-19
> > **Response to Reviewer LKDX (2/2)**
> >
> > Q6. **Section 3.1 could be improved by adding a bit more intuition about the equations.**
> > * Thank you for the suggestion. We have added more intuition in Section 3.1. There are more intuitions at the beginning of Section 3 (page 3). Hope that also helps.
> >
> > Q7. **How many seeds do you use for the standard deviations?**
> > * We use 5 seeds to calculate std. To make it more obvious, we added the seed number into the caption of Table 1 and Table 2 (Page 9).
> >
> > Q8. **Typos and minor revisions.**
> > * We have fixed the typos and made corresponding revisions based on your suggestions.
> > * For 'sparse labels' in P8, we realize this is not a proper word. We use 'few labels' instead, w.r.t. masks with only a few pixels that have labels.
> > * We changed $\mathbb{E}Y$ to $\mathbb{E}[Y]$.
> > * For $N_S$, it is defined in section 3.1 when it first shows up.
> >
> > Thank you again for your detailed review. We hope we have addressed all your concerns. If you have any questions, please kindly let us know. We are happy to discuss further.
> >
> > [1] Monteiro, Miguel, et al. "Stochastic segmentation networks: Modelling spatially correlated aleatoric uncertainty." Advances in Neural Information Processing Systems 33 (2020): 12756-12767.

---

> > > ### Author Response · Authors · 2022-11-24
> > > **Looking forward to further discussion**
> > >
> > > Dear Reviewer LKDX,
> > >
> > > We hope our discussions and updated manuscript have addressed your concerns. Kindly do let us know if there is anything you would like to discuss further. We remain attentive to your feedback.
> > >
> > > Thank you,
> > >
> > > Authors of Paper3111

---

> > > > ### Author Response · Authors · 2022-11-30
> > > > **Looking forward to further discussion**
> > > >
> > > > Dear Reviewer LKDX,
> > > >
> > > > We would like to know whether our responses clarify the questions raised in your initial reviews. We are happy to provide any further discussion.
> > > >
> > > > Thank you,
> > > >
> > > > Paper3111 Authors

---

> > > > > ### Comment · Reviewer_LKDX · 2022-12-12
> > > > > **Thanks for the clarification**
> > > > >
> > > > > Dear authors,
> > > > >
> > > > > thanks a lot for the thorough rebuttal, clarifications, and additional experiments. I am happy to see that the method still achieves good performance on LIDC and on data with inner holes.
> > > > >
> > > > > I have no further questions at this point.
> > > > >
> > > > > I have updated my score to 6 to reflect this.
> > > > >
> > > > > However, one suggestion would be to extend some of the figure captions in the appendix for ease of reading if this paper is accepted.
> > > > >
> > > > > Best, Reviewer LKDX

---

> > > > > > ### Author Response · Authors · 2022-12-12
> > > > > > **Thank you very much!**
> > > > > >
> > > > > > Thank you for your support and suggestions. We will polish the figure captions in the final version.
> > > > > >
> > > > > > Best,
> > > > > >
> > > > > > Paper3111 Authors

---

> ### Author Response · Authors · 2022-12-06
> **Your further feedback is much appreciated**
>
> Thank you again for your initial review. We have added new experiments and illustrations to address your major concerns. Specifically, we
>
> 1) clarified our experiment settings and added a new real-world dataset;
>
> 2) showed with both analysis and experiment that our method can address potential holes.
>
> Please kindly let us know if you have further concerns.

---

### Public Comment · ~Sheng_Liu2 · 2022-11-05
**Discussion on a related work**

Dear authors,

The paper is very interesting and deals with a very important problem in segmentation. I also noticed that this paper is similar, to some extent, to our previous paper "Adaptive Early-Learning Correction for Segmentation from Noisy Annotations" published at CVPR 2022. Could you discuss the difference? I briefly read the paper and believed that there are many novelties besides the annotation correction itself. Thank you in advance.

Best,

Sheng

---

> ### Author Response · Authors · 2022-11-19
> **Reply to Sheng**
>
> Thank you for providing an additional reference. Our method focuses on modeling the spatial correlation in noisy labels and correct labels with an estimated spatial bias, while the mentioned paper employs early learning techniques to prevent the DNN from overfitting to noisy labels, like our baseline methods ELR (Arpit et al., 2017) and CDR (Liu et al., 2020). We have discussed the mentioned paper in the related work (Page 2) and also in Section A.3 of Appendix (Page 17).

---

### Author Response · Authors · 2022-11-19
**Summary of Author Response to All the Reviewers**

We appreciate all reviewers for their time and insightful comments. It seems that all the reviewers have agreed on the importance of the problem we aim to solve. We revised the manuscript based on the constructive feedback and suggestions from the reviewers. We have uploaded the revised version to reflect the modifications (highlighted in blue). There are two most common concerns that reviewers hold.

Q1. **The synthetic experiments may not be able to generalize to other types of noises that are not generated by the proposed noise model. (Reviewer LKDX, VpVv)**
* We would like to clarify that apart from noisy settings ($S_E$ and $S_S$) generated by the proposed noise model, we also include another setting $S_M$, which is the random dilation and erosion noise used in previous methods, like QAM (Zhu et al., 2019) and CLE (Zhang et al., 2020b). We will add this to the paper.

Q2. **There are not enough experiments on real-world noise, especially from medical imaging. (Reviewer LKDX, VpVv, VNP8)**
* We are constrained by the lack of public medical datasets with both true and noisy labels. The suggestion from Reviewer LKDX is brilliant: turning the LIDC-IDRI multi-annotator dataset into a label noise setting. Following the suggestion, we provide an additional experiment: we use the majority voting of different annotations as the true labels and the union of all annotations as noisy labels. The results are reported in the table below. It can be observed that our method outperforms all baselines. Experimental details (Section A.2.1, Page 15) and the table (Table 2, Page 9) have been added to the paper.
| GCE | SCE | CT+ | ELR | CDR | QAM | CLE | SC(Ours) |
| :---: | :---: | :---: | :---: | :---: | :---: | :---: | :---: |
| $42.56\pm0.58$ | $50.40\pm0.15$ | $40.25 \pm 0.11$ | $49.76 \pm 0.40$ | $49.41\pm0.79$ | $43.56 \pm 0.29$ | $49.79 \pm 0.19$ | **53.72** $\pm$ **1.70** |

**Summary of major changes to the manuscript.**

1. We added another real-world dataset LIDC_IDRI in Section 4.1 and provided the results in Table 2.
2. We revised the caption of Figure 3 to make the figure easier to understand.
3. We moved the detailed settings of datasets to Section A.2.1 of Appendix due to page limit.
4. We added two illustrative experiments in Appendix Section A.3. One is to show the issues that previous work has, i.e., they cannot tackle the bias in labels. We also explain why our method is capable of solving this problem. The other one is to show our method still works under labels with potential holes. These are to address concerns from Reviewer LKDX.
5. Following suggestions from reviewers, we fixed typos and reformalized some notations to improve our manuscript. All changes have been colored blue in the updated manuscript.

Below we will address specific concerns from each reviewer.

---

### Decision · Program_Chairs · 2023-01-20

**Decision:**

Accept: poster

**Justification For Why Not Higher Score:**

The reviewer scores do not support it.

**Justification For Why Not Lower Score:**

All reviewers suggest acceptance.

**Metareview: Summary, Strengths And Weaknesses:**

Summary:
This paper proposes an approach to correct for noisy image segmentation annotations by utilizing spatially-aware operations at the segmentation boundary.

Strengths:
- Well written paper
- Novel, simple method that produces improved performance
- Good experimental validation

Weaknesses:
- Limitations of and bias in experimental validation could be discussed more thoroughly
- While the paper handles noisy labels, most labels used in the experiments have synthetic noise. More real data experiments would improve our ability to conclude from the results
- Code is not provided -- although authors promise to release it if accepted

**Note From Pc:**

if the above contains the word "oral" or "spotlight" please see: "oral" presentation means -> notable-top-5% and "spotlight" means -> notable-top-25%. As stated in our emails, we are disassociating presentation type from AC recommendations